# Greenland liquid water discharge from 1958 through 2019

Kenneth D. Mankoff[1], Brice Noël[5], Xavier Fettweis[2], Andreas P. Ahlstrøm[1], William Colgan[1], Ken Kondo[3], Kirsty Langley[4], Shin Sugiyama[3], Dirk van As[1], and Robert S. Fausto[1]

[1]Department of Glaciology and Climate, Geological Survey of Denmark and Greenland (GEUS), Copenhagen, Denmark
[2]Department of Geography, University of Liège, Belgium
[3]Institute of Low Temperature Science, Hokkaido University, Japan
[4]Asiaq-Greenland Survey, Nuuk, Greenland
[5]Institute for Marine and Atmospheric Research, Utrecht University, The Netherlands

**Correspondence:** Ken Mankoff (kdm@geus.dk)

**Abstract.** Greenland runoff, from ice mass loss and increasing rainfall, is increasing. That runoff, as discharge, impacts the physical, chemical, and biological properties of the adjacent fjords. However, where and when the discharge occurs is not readily available in an open database. Here we provide datasets of high-resolution Greenland hydrologic outlets, basins, and streams, and a daily 1958 through 2019 time series of Greenland liquid water discharge for each outlet. The data include $24507$ ice marginal outlets and upstream basins, and $29635$ land coast outlets and upstream basins, derived from the 100 m ArcticDEM and 150 m BedMachine. At each outlet there are daily discharge data for $22645$ days - ice sheet runoff routed subglacially to ice margin outlets, and land runoff routed to coast outlets - from two regional climate models (RCMs; MAR and RACMO). Our sensitivity study of how outlet location changes for every inland cell based on subglacial routing assumptions, shows that most inland cells where runoff occurs are not highly sensitive to those routing assumptions, and outflow location does not move far. We compare RCM results with 10 gauges from streams with discharge rates spanning four orders of magnitude. Results show that for daily discharge at individual basin scale the 5 to 95 % prediction interval between modeled discharge and observations generally falls within plus-or-minus a factor of five (half an order of magnitude, or +500%/-80%). Results from this study are available at doi:10.22008/promice/freshwater (Mankoff, 2020a) and code is available at http://github.com/mankoff/freshwater (Mankoff, 2020b).

## 1 Introduction

Over the past decades, liquid runoff from Greenland has increased (Mernild and Liston, 2012; Bamber et al., 2018; Trusel et al., 2018; Perner et al., 2019) contributing to mass decrease (Sasgen et al., 2020). When that runoff leaves the ice sheet and discharges into fjords and coastal seas, it influences a wide range of physical (Straneo et al., 2011; An et al., 2012; Mortensen et al., 2013; Bendtsen et al., 2015; Cowton et al., 2015; Mankoff et al., 2016; Fried et al., 2019; Cowton et al., 2019; Beckmann et al., 2019), chemical (Kanna et al., 2018; Balmonte et al., 2019), and biological (Kamenos et al., 2012; Kanna et al., 2018; Balmonte et al., 2019) systems (Catania et al., 2019). The scales of the impacts range from instantaneous at the ice-ocean boundary to decadal in the distal ocean (Gillard et al., 2016). The influence of freshwater on multiple domains and disciplines (Catania et al., 2019) is the reason several past studies have estimated runoff and discharge at various temporal and spatial

scales (e.g. Mernild et al. (2008, 2009, 2010a); Langen et al. (2015); Ahlstrøm et al. (2017); Citterio et al. (2017); van As et al. (2018); Bamber et al. (2018); Perner et al. (2019); Slater et al. (2019)).

To date no product provides discharge estimates at high spatial resolution (~100 m; resolving individual streams), daily temporal resolution, for all of Greenland, covering a broad time span (1958 through 2019), from multiple regional climate models (RCMs), and with a simple database access software to support downstream users. Here we present these data. In the following description and methods, we document the inputs, assumptions, methodologies, and results we use to estimate Greenland discharge from 1958 through 2019.

Freshwater discharge from Greenland primarily takes three forms: solid ice from calving at marine terminating glaciers, submarine meltwater from ice-ocean boundary melting at marine terminating glaciers, and liquid runoff from melted inland surface ice, rain, and condensation. A recent paper by Mankoff et al. (2020) targets the solid ice discharge plus submarine melt budget by estimating the ice flow rate across gates 5 km upstream from all fast-flowing marine terminating glaciers in Greenland. Complementing that paper, this paper targets Greenland's point-source liquid water discharge budget by partitioning RCM runoff estimates to all ice margin and coastal outlets. The sum of these data and Mankoff et al. (2020) is an estimate of the majority of freshwater (in both liquid and solid form) volume flow rates into Greenland fjords. Those two terms comprise the bulk but not all freshwater - they exclude precipitation directly onto the fjord or ocean surface, and relatively minor contributions from evaporation and condensation, sea ice formation and melt, or subglacial basal melting.

## 2 Input and validation data

### 2.1 Static data

The static products (streams, outlets, and basins (Fig. 1)) are derived from an ice-sheet surface digital elevation model (DEM), an ice sheet bed DEM, an ice-sheet mask, the land surface DEM, and an ocean mask. For the surface DEM, we use ArcticDEM v7 100 m (Porter et al., 2018). Subglacial routing uses ArcticDEM and ice thickness from BedMachine v3 (Morlighem et al., 2017a, b). Both DEMs are referenced to the WGS84 ellipsoid. For the ice mask we use the Programme for Monitoring of the Greenland Ice Sheet (PROMICE) Ice Extent (Citterio and Ahlstrøm, 2013). For the ocean mask we use the Making Earth System Data Records for Use in Research Environments (MEaSUREs) Greenland Ice Mapping Project (GIMP) Land Ice and Ocean Classification Mask, Version 1 (Howat, 2017b; Howat et al., 2014).

### 2.2 RCM time series

The time series product (daily discharge) is derived from daily runoff estimates from RCM calculations over the land and ice areas of Greenland. We use the Modèle Atmosphérique Régional (MAR; Fettweis et al. (2017)) and the Regional Atmospheric Climate Model (RACMO; Noël et al. (2019)). Runoff, $R$, is defined by

$$R = ME + RA - RT - RF. \tag{1}$$

In Eq. 1, $ME$ is melt, $RA$ is rainfall, $RT$ is retention, and $RF$ is refreezing. In RACMO, retention occurs only when firn is present (not with bare ice). MAR does have a delay for bare ice runoff. Neither have a delay for land runoff. Both RCM outputs were provided regridded to the same 1 km grid using an offline statistical down-scaling technique based on local vertical runoff gradient applied to the sub-grid topography (Noël et al., 2016; Fettweis et al., 2020). MAR (v 3.11; Delhasse et al. (2019)) ran with 7.5 km resolution and ERA5 6-hour forcing. RACMO (v 2.3p2; Noël et al. (2018, 2019)) ran with 5.5 km resolution and ERA-Interim 6-hour forcing. Runoff is assigned an uncertainty of ±15 % (Sect. 4.3.3).

## 2.3 River discharge observations

We use 10 river discharge daily time series to validate the results of this work. The name, location, time coverage, and relevant data and scientific publications associated with each of these observational data are listed in Table 1.

**Table 1.** Table of observation locations, time spans, and associated references. Coordinates are decimal degree W and N.

| Location | Lon | Lat | Time | Data | Publication | Fig(s). |
|---|---|---|---|---|---|---|
| Kiattuut Sermiat | 45.33 | 61.21 | 2013 | Hawkings et al. (2016a) | Hawkings et al. (2016b) | 1 3 4 5 10 |
| Kingigtorssuaq (Nuuk) | 51.5801 | 64.1387 | 2008-2018 | Langley (2020) | | 1 3 4 11 |
| Kobbefjord (Nuuk) | 51.3810 | 64.1336 | 2006-2017 | Langley (2020) | | 1 3 4 14 |
| Leverett Glacier | 50.17 | 67.06 | 2009-2012 | Tedstone et al. (2017) | Hawkings et al. (2015) | 1 3 4 5 9 |
| Oriartorfik (Nuuk) | 51.4066 | 64.1707 | 2007-2018 | Langley (2020) | | 1 3 4 12 |
| Qaanaaq | 69.3030 | 77.4753 | 2017-2018 | Kondo and Sugiyama (2020) | Sugiyama et al. (2014) | 1 3 4 5 17 |
| Røde Elv (Disko) | 53.4989 | 69.2534 | 2017 | Langley (2020) | | 1 3 4 5 6 15 |
| Teqinngalip (Nuuk) | 51.5484 | 64.1586 | 2007-2018 | Langley (2020) | | 1 3 4 13 |
| Watson River | 50.68 | 67.01 | 2006-2019 | van As et al. (2018) | van As et al. (2018) | 1 3 4 5 7 8 |
| Zackenberg | 20.5628 | 74.4722 | 1996-2018 | Langley (2020) | | 1 3 4 5 16 |

## 3 Methods

### 3.1 Terminology

We use the following terminology throughout the document:

- Runoff refers to the unmodified RCM data products - melted ice, rain, condensation, and evaporation that comprise the RCM runoff output variable.

- Discharge refers to the runoff after it has been processed by this work - routed to and aggregated at the outlets. Depending on context, discharge may also refer to the observed stream discharge (Table 1).

- Basins refer to the 100 m x 100 m gridded basins derived from a combination of the ArcticDEM product and the mask.

- Mask refers to the surface classification on that 100 m x 100 m grid and is one of ice, land, or ocean (also called fjord or water). When referring to the surface classification in the RCM, we explicitly state "RCM mask".

- MAR and RACMO refer to the RCMs, but when comparing discharge estimates between them or to observations, we use MAR and RACMO to refer to our discharge product derived from the MAR and RACMO RCM runoff variables, rather than repeatedly explicitly stating "discharged derived from [MAR|RACMO] runoff". The use should be clear from context.

## 3.2 Streams, outlets, and basins

Streams are calculated from the hydraulic head $h$ which is the DEM surface for land surface routing, or the subglacial pressure head elevation for subglacial routing. $h$ is defined as

$$h = z_b + k\frac{\rho_i}{\rho_w}(z_s - z_b), \tag{2}$$

with $z_b$ the ice-free land surface and basal topography, $k$ the flotation fraction, $\rho_i$ the density of ice (917 kg m$^{-3}$), $\rho_w$ the density of water (1000 kg m$^{-3}$), and $z_s$ the land surface for both ice free and ice covered surfaces.

Eq. 2 comes from Shreve (1972) where the hydropotential has units Pa, but here is divided by gravitational acceleration $g$ times the density of water $\rho_w$ to convert the units from Pa to m. We compute $h$ and from that streams, outlets, basins, and runoff for a range of subglacial pressures, implemented as a range of $k$ values: 0.8, 0.9, and 1.0. We use these three scenarios to estimate sensitivity of the outlet location for all upstream cells, but otherwise only use results from the $k = 1.0$ scenario. Eq. 2 makes the assumption that when ice is present all water routes subglacially, meaning that water flows from the surface to the bed in the grid cell where it is generated. In reality, internal catchments and moulins likely drain waters to the bed within a few km of their source (Yang and Smith, 2016). The difference between some supraglacial flow and immediate subglacial flow is not likely to impact results because discharge is reported only at the outlet locations.

We use the GRASS GIS software (Neteler et al., 2012; GRASS Development Team, 2018) and the `r.stream.extract` command configured for single-flow direction from eight neighbors (SFD-8) to calculate streams and outlets at the ice edge and coast. Streams are defined only if their upstream contributing area is above a threshold (> 3 km$^2$), so small basins may have outlets but no streams. The software fills all sinks so that all water flows to the domain edge. We then use the `r.stream.basins` tool (Jasiewicz and Metz, 2011) to calculate basins upstream from each outlet. Basins < 1 km$^2$ are absorbed into their largest neighbor and the associated outlets are dropped.

### 3.2.1 Outlet sensitivity

The three choices of $k$ generate three scenarios of basins and outlets, and we use this to show sensitivity of every ice grid cell to these choices. After three $k$-scenarios, each cell has three possible outlets, where each outlet is an (x,y) coordinate. To show results in a map view, we reduced these six properties (three 2D coordinates) to a single property. For every grid cell in the

ice domain we compute the distance between each outlet and the other two (six becomes three), and then select the maximum (three becomes one). Fig. 2 displays the maximum distance - a worst-case scenario - of how far the outlet of every inland ice cell may move due to basal routing assumptions.

## 3.3 Discharge and RCM coverage

RCM runoff is summed over each basin for each day of RCM data, and assigned to each outlet for that day. This assumes routing between the runoff and the outlet is instantaneous, so all analyses done here include a seven-day smooth applied to the RCM discharge product (cf. van As et al. (2017)). The released data do not include any smoothing.

The alignment of the RCM and the basins do not always agree. Each 100 m x 100 m ArcticDEM pixel is classified as ice (Citterio and Ahlstrøm, 2013), ocean (Howat, 2017b), or land (defined as neither ice nor ocean). However, the classification of the mask cells and the 1 km$^2$ RCM domains do not always agree - for example, when a mask cell is classified as ice but the matching RCM cell is land. This disagreement occurs almost everywhere along the ice margin because the 1 km RCM boundary and the 100 m mask boundary rarely perfectly align. The ice margin is where most runoff occurs per unit area due to the highest temperatures at the lowest ice elevations, so small changes in masks in these locations can introduce large changes in RCM outputs.

We adjust for this imprecise overlap and scale the RCM results to the basin area. Where the mask reports ice and a RCM reports land, the RCM land runoff fraction is discarded, and the RCM ice runoff fraction over this basin is adjusted for the uncovered basin cells (and vice versa for basin land and RCM ice). Small basins with no RCM coverage of the same type have no runoff.

Runoff adjustments using this method are underestimated for large basins with large inland high elevation regions with low runoff, because this method fills in misaligned cells with each days average discharge, but the misalignment (missing runoff) occurs at the ice sheet edge where maximum runoff occurs. However, given that the basin is large, misalignment is proportionally small, and therefore errors are proportionally small. Conversely, when misalignment is proportionally large (e.g. a basin is only 1 % covered by the same RCM classification), this implies a small basin. Because the basin is small, the covered region (no matter how much smaller) must be nearby and not climatically different.

RCM inputs are also scaled to adjust for the EPSG:3413 non-equal-area projection. This error is up to 8 % for some grid cells, but ranges from - 6 % to + 8 % over Greenland and the cumulative error for the entire ice sheet is < 8 %.

## 3.4 Validation

We validate the modeled outlet discharge against the observations first in bulk and then individually. Bulk comparisons are done with scatter plots (Figs. 3 & 4), and modified Tukey plots comparing observations vs. the ratio of the RCMs to observations (Fig. 5, based on Tukey mean-difference plots, also known as Bland-Altman plots (Altman and Bland, 1983; Martin Bland and Altman, 1986)).

We introduce the graphics here as part of the methods to reduce replication in figure captions - we show 10 nearly identical graphics (Figs. 7 and 9 through 17) for 10 different observation locations, and each graphic uses the same template of six panels.

For each figure (Figs. 7, 9 to 17), the top panel (a) shows a satellite basemap with the land portion of the basin of interest (if it exists) outlined in dark green, the streams within that basin in light green, the basin outlet as an orange filled diamond, and the stream gauge location as an orange unfilled diamond. Ice basin(s) that drain to the outlet are outlined in thick dark blue if they exist, and all other ice basins in thin dark blue. Both MAR and RACMO use the same domains. The RCM ice domain is in light blue, and the RCM land domain is not shown, but is outside the light blue ice domain (not including the water). The scale of each map varies, but the basins lines (green and dark blue) are on a 100 m grid, and the RCM grid cells (light blue) are on a 1 km grid.

Panel b shows an example time series - whatever data are available for the last calendar year of the observations.

Panel c shows a scatter plot of observations vs. RCM-derived discharge. This is the same data shown in Fig. 3, but subset to just the basin of interest. Color encodes day-of-year, and a kernel density estimation (KDE) of the discharge highlights where most points occur - not necessarily visible without the KDE because the points overlap (total number of plotted points is printed on the graphic near "n:"). The $r^2$ correlation coefficient for each RCM-derived discharge is displayed. The gray band shows the 5 to 95 % prediction interval, and the three solid lines mark the 1:1, 1:5, and 5:1 ratios.

Panel d shows observations vs. the ratio of the RCM to the observations. This is the same data shown in Fig. 5, but subset to just the basin of interest. Color denotes sample density (similar to the KDE in panel c). The horizontal lines mark the mean, 0.05, and 0.95 quantile of the ration between the RCM and the observations. A value of 1 (or $10^0$) is agreement between observations and the RCM, and a value of 2 or 0.5 is a factor of 2 or a +100/-50 % disagreement. The horizontal split marks the bottom 1/3rd and top 2/3rds quantiles of discharge.

## 4    Product evaluation and assessment

Results of this work include 1) ice-margin terminating streams, outlets, and basins, 2) coast-terminating streams, outlets, and basins, 3) discharge at the ice-marginal outlets from ice runoff and 4) discharge at the coastal outlets from land runoff. Discharge products are provided from both the MAR and RACMO RCMs. We note that our subglacial streams represent where the model routes the water, and does not indicate actual streams, unlike the land streams that do appear near actual streams when compared to satellite imagery. Even so, these streams routed using simple subglacial theory show remarkable alignment with ice surface streams and lakes visible in satellite imagery. This may support the theory that basal topography exerts a strong control on supraglacial hydrology (Lampkin and VanderBerg, 2011; Sergienko, 2013; Crozier et al., 2018), or may indicate a poorly represented and smooth bed in BedMachine, and therefore Eq. 2 is effectively applying surface routing in these locations.

Of the 361,950 km$^2$ of basin land cells, the RCMs cover 339,749 km$^2$ (~94 %) with their land grid cells, and 22,201 km$^2$ (~6 %) of basin grid cells are filled in with our coverage algorithm (Sect. 3.3; the RCMs have these as ice or ocean). 51,532 km$^2$

of RCM land are discarded because the basins classify part or all of these cells as ice or ocean. Of the 1,781,816 km$^2$ of basin ice cells, the RCMs cover 1,760,912 km$^2$ (~99 %) with their ice cells, and 20,904 km$^2$ (~1 %) of basin grid cells are filled in (the RCMs have these as land or ocean). 21,793 km$^2$ of RCM ice are discarded, because the basins classify part or all of these cells as land or ice (Table and data available in Supplemental Online Material).

Our coverage correction (Sect. 3.3) adjusts RCM ice runoff values by ~3 %. Discarding RCM ice runoff that does not match the underlying mask ice cells results in a 5 % reduction in discharge. However, applying our coverage algorithm to adjust RCM inputs for regions where basins have ice but the RCMs do not results in an 8 % increase from the reduced discharge (net gain of ~3 %). A similar adjustment occurs for RCM land runoff.

## 4.1 Comparison with previous similar work

Our static products - streams, outlets, and basins - have been previously estimated. Lewis and Smith (2009) identified 293 distinct hydrologic ice basins and provided a data set of ice basins and ice margin outlets. Our work, a decade later, has ~2 orders of magnitude more basins and outlets because of the higher resolution of the input data, and includes additional data. We provide ice basins, ice margin outlets, ice streams with metadata, land basins, coastal outlets, and land streams with metadata. Lewis and Smith (2009) generated basins from a 5 km DEM, compared to the 100 m DEM used here. Routing with a 5 km DEM that does not capture small-scale topography is likely to cause some basins and outlets to drain into an incorrect fjord - we find that some land basins delineated with even the 150 m BedMachine land surface may drain into the incorrect fjord, but we did not find similar errors with the 100 m ArcticDEM product used in this work.

Our time-series product - discharge, also has existing similar products. The most recent of these is from Bamber et al. (2018) who provide a data product at lower spatial resolution (5 km), lower temporal resolution (monthly), and only coastal discharge, not coastal basins, ice basins, or ice margin outlets and discharge. However, Bamber et al. (2018) surpasses our product in that spatial coverage includes a larger portion of the Arctic including Iceland, Svalbard, and Arctic Canada. Furthermore, by providing data at 5 km spatial and monthly temporal resolution, Bamber et al. (2018) implements the main strategy suggested here to increase the signal-to-noise ratio of the data - averaging discharge in space or time (see Sect. 4.3.5).

We show both the geospatial and temporal differences between this product and the Bamber et al. (2018) for an example location - Disko Island (Fig. 6). Spatially our product allows assessment of discharge at interior locations, necessary when comparing with observations that are not at the coast (for example, the Leverett Glacier observations (Fig. 9)). Temporally, the MAR and RACMO runoff summed over all of Disko Island and to monthly resolution is similar to the monthly Disko discharge of Bamber et al. (2018), but the daily resolution shows increased variability and individual discharge events (from warm days or rain) not seen in the monthly view.

A similar GIS workflow was presented by Pitcher et al. (2016) only focusing on the discharge uncertainty from basal routing assumptions (the $k$ parameter in Eq 2). We find these differences to be smaller than the differences between RCMs or between RCM and observations (see Sect. 4.3).

## 4.2 Validation against observations

Here we compare our results to all publicly accessible observations we could find, or willing to become open and publicly accessible as part of this work (Table 1).

This validation compares observations with discharge at stream gauges derived from RCM runoff estimates, much of it coming from far inland on the ice sheet. Disagreement is expected and does not indicated any specific issues in the RCMs, but are instead likely due to our routing algorithm (Sect. 3.3).

Below we discuss first the validation for all discharge estimates together, then the individual outlets. For the individual outlets we begin by focusing on the "problematic" results in order of severity: Watson River (Figs. 7 & 8), Leverett Glacier (Fig. 9), and Kiattuut Sermiat (Fig. 10), and show that for two of these three, simple solutions are available, although manual intervention is needed to detect the issue and then adjust results.

### 4.2.1 Bulk validation

A comparison of every day of observational data with discharge > 0 (15,778 days) and the two RCMs (Fig. 3) shows good agreement with $r^2$ of 0.45 and 0.88 for discharge derived from MAR and RACMO runoff respectively (hereafter "MAR" and "RACMO"). This comparison covers more than four orders of magnitude of modeled and observed discharge. The RACMO vs. observed discharge is within a factor of five (e.g. plus-or-minus half an order of magnitude), although both RCMs report only ~50 % of the observed discharge for the largest volumes at the Watson River outlet (Fig. 7). The reason for the disagreement at the Watson River outlet is discussed in detail in Sect. 4.2.2.

The four near-Nuuk GEM basins (Table 1, Sect. 4.2.5) have ice basins but either no or limited coverage in the RCMs. When excluding these basins from the comparison the $r^2$ agreement changes to 0.59 and 0.78 for MAR and RACMO respectively and the 5 to 95 % prediction interval is significantly smaller for MAR (red band in Fig. 3). The largest disagreements throughout this work comes from these small basins with no RCM coverage. These disagreements are therefore indicative of differences between the land/ice classification mask used by the RCMs compared with the basin masks used here, not necessarily an insufficient ability of the models to simulate melting (near-surface) climate conditions.

Fig. 4 shows a similar view as Fig. 3, but here each observational data set and associated daily discharge is summed by year for the days in that year that observations exist (hence units $m^3$ and not $m^3$ $yr^{-1}$; for example four "L" means there are four calendar years with some observations at the Leverett outlet). Here it is more clear that the Watson River outlet (Sect. 4.2.2) reports ~50 % of the observed discharge, the Kiattuut Sermiat outlet (Sect. 4.2.4) over-estimates discharge, and the remainder fall within the factor-of-two lines, except for low discharge at Kingigtorssuaq in the MAR RCM where the RCMs do not cover that small glacier (Sect. 4.2.5).

Because discharge spans a wide range (~4 orders of magnitude, Fig. 3), a high correlation ($r^2$ of 0.88, Fig. 3) may be due primarily to the range which is larger than the error (Altman and Bland, 1983; Martin Bland and Altman, 1986). Fig. 5 compensates for this by comparing the observations with the ratio of the RCM to the observations. This graphic again excludes the four near-Nuuk GEM basins. From Fig. 5, the top 2/3rds of observed discharge has modeled discharge under-estimated by

230 a factor of 0.78 (MAR) and 0.73 (RACMO), and 5 to 95 % quantile of 0.30 to 2.08. The top 2/3rds of observed discharge spans ~2 orders of magnitude (width of horizontal lines, from ~$10^1$ to ~$10^3$ $m^3$ $s^{-1}$). The ratio of the RCMs to these observations for the top 2/3rds has a 5 to 95 % quantile range of ~1 order of magnitude (distance between horizontal lines, from $\log_{10} 0.3$ to $\log_{10} 2.08 = 0.84$). The 5 to 95 % quantile range of the ratio between the RCMs and the observations is therefore half the range of the observations. Put differently, days with high observed discharge may have modeled discharge within ±0.5 order

of magnitude, or plus-or-minus a factor of five, or +500/-80 %. The modeled discharge is not likely to move farther than this from the observations, and high discharge remains high.

The bottom third of discharge is where the largest disagreement occurs. The mean model values are near the observed - the ratio of RCM to observed discharge is scaled by 0.67 for MAR (~33 % low) and 1.08 for RACMO (~8 % high), but the 5 to 95 % quantile range of the ratio between RCM and observations is large. Although large uncertainties for low discharge may not

seem to matter for some uses (e.g. estimates of total discharge from Greenland, which is dominated by the largest quantities of discharge), it may matter for other uses. The bottom 1/3 quantile of observed discharge spans 3 orders of magnitude ($10^{-2}$ to ~$10^1$) but the uncertainty of the RCM to observations ratio spans ~4 and ~2 orders of magnitude for MAR and RACMO respectively (~$10^{-3}$ to ~$2.2 \times 10^1$ MAR; ~$10^{-1}$ to $2.2 \times 10^1$ RACMO).

### 4.2.2   Watson River

The Watson River discharge basin area is 1882 $km^2$, of which 521 $km^2$ (28 %) are land and 1361 $km^2$ (72 %) are ice (Fig 7a). The partial (last calendar year) discharge time series shows MAR and RACMO agree well with each other, but have a maximum of 500 $m^3$ $s^{-1}$ while observations are up to 4x more (Fig. 7b). Low discharge (both early and late season) is over-estimated and high discharge is under-estimated, approximately equal for both RCMs (Fig. 7c). The low discharge over-estimate ranges from a mean multiple of 1.68 (MAR) and 1.57 (RACMO) to a +95 % quantile ratio of ~70 (MAR) and ~52 (RACMO). The high-

discharge under-estimate has a mean multiple of ~0.5 for both MAR and RACMO, and a 5 to 95 quantile range of between 0.23 to 1.09.

The Watson River discharge presented here is approximately half of the van As et al. (2018) discharge for high discharge. The large underestimate for high discharge may be due to either errors in the basin delineation used in this study, errors in the stage-discharge relationship used by van As et al. (2018), errors in the RCM runoff estimates, or a combination of the

255 above three. All three of these error sources increase with high discharge (and associated melt): Basin delineation becomes less certain with inland distance from the ice sheet margin. The river stage-discharge conversion becomes less certain at high stage levels. Runoff calculations become less certain from a snow surface than an ice surface, because of e.g. snow density, subsurface refreezing, and surface darkening.

The complexity of estimating the area of the Watson River catchment is described by Monteban et al. (2020), who note that

previous studies have used values ranging from 6131 $km^2$ (Mernild et al., 2010b) to 12547 $km^2$ (van As et al., 2012). Our basin is smaller than the basin used in van As et al. (2018) and similar to Mernild et al. (2018) who attributed the difference between their modeled outflow and observations from van As et al. (2017) to their decision to use surface rather than subglacial routing, and applied a correction term. We find that our basin does not include a separate basin to the south that is part of the Watson

River ice basin in van As et al. (2018) (from Lindbäck et al. (2015) and Lindbäck et al. (2014)). We are able to recreate the van As et al. (2018) basin but only when using the Lindbäck et al. (2014) bed and the Bamber et al. (2013) surface. When using any other combination of bed DEM, surface DEM, or $k$ values, we are unable to match the Lindbäck et al. (2015) basin. Instead all our basins resemble the basin shown in Fig 7. To solve this, we manually select two large ice basins to the south of the Watson River ice basin. Modeled and observed discharge agree after including these two basins (Fig. 8), suggesting basin delineation, not stage-discharge or RCM runoff is the primary cause for this disagreement. Furthermore, it is the additional width at lower elevation from the larger basin, not the increased inland high-elevation area, that likely contributes the runoff needed to match the observations, because 85 % of all surface runoff occurs below 1350 m, and almost all below 1850 van As et al. (2017).

At the Watson River outlet, there is no reason to suspect this product underestimates observed discharge by 50 %. The observations are needed to highlight the disagreement. Once this disagreement is apparent, it is also not clear what to do to reduce the disagreement without the previous efforts by Lindbäck et al. (2015) and Lindbäck et al. (2014). Basin delineation is discussed in more detail in the uncertainty section (Sect. 4.3.2). The other two "problematic" areas highlighted above (Sect. 4.2) can be detected and improved without observational support.

### 4.2.3 Leverett Glacier

The Leverett Glacier basin area is 1361 km$^2$ and 100 % ice (Fig 9a). The partial (last calendar year) discharge time series shows MAR and RACMO agree well with each other and with the observations (Fig. 9b), with no seasonal dependence (Fig 9c). The 5 to 95 % prediction interval for MAR is generally within the 1:5 and 5:1 bands, with a larger spread for RACMO (Fig 9c). High model discharge is 3 % higher than observed (MAR) or 25 % higher than observed (RACMO), and the 5 to 95 quantile range of the ratio is between 0.73 and 1.62 (MAR) and 0.83 and 2.02 (RACMO). Low model discharge is also centered near the observations, but as always larger errors exist for low discharge (Fig 9d).

This basin is problematic because the basin feeding the outlet is small (< 5 km$^2$), but even without the observational record satellite imagery shows a large river discharging from the ice sheet here. Meanwhile, a large (100s of km$^2$) ice basin does discharge just a few 100 m away, but not upstream of this gauge location. We therefore adjust the gauge location onto the ice (equivalent to selecting a different outlet) so that our database access software selects what appears to be the correct basin given the size of the stream in the satellite imagery (Fig. 9).

The plots shown here use the adjusted gauge location and modeled discharge appears to match the observed discharge. When plotting (not shown) the modeled discharge for the outlet just upstream of the true gauge location, results are clearly incorrect. This issue - small basins at the margin and incorrect outlet location - is persistent throughout this product and discussed in more detail in Sect. 4.3.2.

The Leverett Glacier basin is a subset of the Watson River outlet basin (Sect. 4.2.2). The strong agreement here supports our claim that the Watson River disagreement is not from the RCM runoff or the stage-discharge relationship, but more likely due to basin area. The correct Watson River basin should include some basins outside of the Leverett Glacier basin that still drain to the Watson River outlet gauge location.

#### 4.2.4 Kiattuut Sermiat

The Kiattuut Sermiat discharge basin area is 693 km$^2$, of which 391 km$^2$ (56 %) are land and 302 km$^2$ (44 %) are ice. The basin area is incorrectly large because the land basin reported and shown includes the entire basin that contains the discharge point, of which some is downstream (Fig 10a). However, only ~25 % of runoff comes from the land, and only a small portion of the land basin is downstream of the gauge location, so this is not enough to explain the discharge vs. observation disagreement. The partial (last calendar year) discharge time series shows MAR and RACMO agree well with each other, but are significantly higher than the observations (Fig. 10b). Both low and high discharge are over-estimated, but the 5 to 95 % quantile range of the ratio are within a factor of five (Fig 10c), with a mean ratio between 1.71 (RACMO bottom 1/3rd of discharge) to 2.44 (MAR high 2/3rds discharge)

The Kiattuut Sermiat gauge is in a problematic location in terms of determining the actual (non-theoretical) upstream contributing area. Similar to the Leverett Glacier gauge location, the issues here can be estimated independent of observational data. Specifically, it is not clear if this stream includes water from the larger glacier to the east and east-northeast that feeds this glacier (Fig. 10a) - in our delineation it does not. Furthermore, several glaciers to the north-northwest and detached from the glacier near the stream gauge appear to drain into a lake that then drains under the glacier and then to the stream gauge. This latter issue is observable in any cloud-free satellite imagery and does not need the basin delineations provided here to highlight the complexities of this field site. Nonetheless, RCM discharge estimates are only slightly more than double the observations.

The Kiattuut Sermiat gauge location may have been selected in part due to its accessibility - it is walking distance from the Narsarsuaq airport. The data may also suit their intended purpose well and there are likely many results that can be derived independent of the area or location of the upstream source water. However, if the location or area of the upstream contributions are important, then gauge location should balance ease of access and maintenance with the ease with which the data can be interpreted in the broader environment.

#### 4.2.5 GEM observations near Nuuk

Four Greenland Ecosystem Monitoring Programme (GEM) stream gauges are located near Nuuk with similar basin properties. All are small (7.56 to 37.52 km$^2$), and 10 % to 25 % ice in the basin mask, but two of the four (Kingigtorssuaq (Fig. 11) and Oriartorfik (Fig. 12)) contain small glaciers contributing to observed discharge but no RCM ice cells cover those glaciers, and the remaining two (Teqinngalip (Fig. 13) and Kobbefjord (Fig. 14)) have several small glaciers, but only one per basin has RCM ice coverage.

All four of these basins show some weak agreement. The maximum r$^2$ is 0.47 (Fig. 13c) and the minimum is 0.11 (Fig 11c), but we note that the worst agreement comes from a basin with no glaciers in the RCM domain, and that in all cases the mean high discharge agrees well, suggesting high discharge in these small basins with few small glaciers may be due to rain (captured in the RCMs) rather than warm days and melted ice.

### 4.2.6 Remaining observations

Three additional stream gauges remain: Røde Elv, Zackenberg, and Qaanaaq.

The Røde Elv basin is situated at the southern edge of Disko Island (Fig. 6). It has an area of 100 km$^2$, of which 72 km$^2$ are land and 28 km$^2$ are ice (Fig 15a). The partial (last calendar year) discharge time series shows MAR, RACMO, and the observations all in approximately the same range but with high variability (Fig. 15b). Of the few samples here (n = 98), most are within the factor-of-five bands for MAR and a few more are outside the bands for RACMO (Fig. 15c). Mean discharge offset ranges from a ratio of 0.82 (RACMO low) to 1.85 (MAR low), with high discharge estimates slightly closer to observations -

a 48 % and 77 % overestimate for MAR and RACMO respectively (Fig. 15d).

    The Zackenberg basin in NE Greenland has an area of 487 km$^2$, of which 378 km$^2$ (78 %) are land and 109 km$^2$ (22 %) are ice (Fig. 16a). The partial (last calendar year) discharge time series shows disagreement between MAR and RACMO that generally bound the observations (Fig. 16b). RACMO-derived discharge is consistently high for low discharge early in the year, but both discharge products fall mostly within the factor-of-five bands (Fig. 16c). For high discharge, mean modeled discharge

is 9 % high (MAR) and 24 % low (RACMO), and has worst-case 5 to 95 % quantile range low by a factor of 0.29 (Fig. 16d).

    The Qaanaaq basin in NW Greenland has an area of 13.2 km$^2$, of which 2.2 km$^2$ (17 %) are land and 11 km$^2$ (83 %) are ice (Fig. 17a). The partial (last calendar year) discharge time series shows disagreement between MAR and RACMO that generally bound the observations (Fig 17b). Of the few samples (n = 82), MAR preferentially over-estimates and RACMO under-estimates discharge, but both generally within a factor of 5 (Fig 17c). The mean high discharge ratio is 1.26 (MAR) and

0.4 (RACMO) from Fig. 17d.

### 4.3 Uncertainty

The volume of data generated here is such that manually examining all of it or editing it to remove artifacts or improve the data would be time and cost prohibitive. A similar warning is provided with the ArcticDEM data used here. However, any ArcticDEM issues interior to a basin do not impact results here that are aggregated by basin and reported at the outlet.

ArcticDEM issues that cross basin boundaries should only impact a small part of the basin near the issue.

    Uncertainty from RCM inputs and observations are considered external to this work, although they are still discussed (Sects. 4.3.3 and 4.3.4). In this work, we introduce one new source of uncertainty - the routing model, which generates both temporal (runoff delay) and spatial (basin delineation) uncertainty.

### 4.3.1 Temporal uncertainty

The RCMs include a time lag between when water melts in the model and when it leaves a grid cell. RACMO retention occurs only when there is firn cover (no retention when bare ice melts); MAR includes a time delay of up to 10 days that is primarily a function of surface slope (Zuo and Oerlemans, 1996; Yang et al., 2019). However, neither model includes a subglacial system. Properly addressing time delays with runoff requires addressing storage and release of water across a variety of timescales in a variety of media: firn (e.g. Munneke et al. (2014); Vandecrux et al. (2019)), supraglacial streams and lakes (e.g. Zuo and

Oerlemans (1996); Smith et al. (2015); Yang et al. (2019)), the subglacial system (e.g. Rennermalm et al. (2013)), possibly terrestrial streams and lakes (e.g. van As et al. (2018)) and a variety of other physical processes that are not within the scope of surface mass balance (SMB) modeling. Runoff delay can be implemented outside the RCMs (e.g. Liston and Mernild (2012); Mernild et al. (2018)), but for this version of the product we assume that once an RCM classifies meltwater as "runoff", it is instantly transported to the outlet. Actual lags between melt and discharge range from hours to years (Colgan et al., 2011; van As et al., 2017; Rennermalm et al., 2013; Livingston et al., 2013).

Data released here includes no additional lag beyond the RCM lag, although a 7-day running mean (cf. van As et al. (2017)) is included in all of the results presented here except Fig. 6 which shows monthly summed data, and Fig. 4 which shows yearly summed data. When increasing the signal to noise by summing by year (Fig. 4 vs. Fig. 3), model results more closely match observations.

### 4.3.2 Basin uncertainty

Basin uncertainty is a function of the subglacial routing assumptions (the $k$ parameter in Eq. 2, which in reality varies in both space and time). However, basin uncertainty does not necessary translate to discharge uncertainty. For example, when comparing two $k$ simulations, a large basin in simulation $k_0$ may change only its outlet by a few grid cells in $k_1$. A small micro-basin may appear in $k_1$ with its outlet in the same grid cell as the large $k_0$ outlet. The large change in discharge between the two outlets at the same location in $k_0$ and $k_1$ is not an appropriate estimate of uncertainty - rather the large basin in $k_0$ should be compared with the almost entirely overlapping large basin in $k_1$ with the different outlet. This fluidity of basins and outlets between $k$-scenarios makes it almost impossible to define, identify, and compare basins between scenarios, unless working manually with individual basins (as we did, for example, at the Leverett Glacier observation location, modeled upstream basin, and adjusted upstream basin (see Sect. 4.2.3)).

Another example has a large basin in simulation $k_0$ and a similarly large basin in simulation $k_1$ draining out of the same grid cell, but overlapping only at the outlet grid cell. Upstream the two do not overlap and occupy different regions of the ice sheet. These two basins sharing one outlet (between different $k$ simulations) could have similar discharge. Put differently, although inland grid cells may change their outlet location by large distances under different routing assumptions (Fig. 2), that does not imply upstream basin area changes under different routing assumptions. Large changes in upstream catchment area are possible (Chu et al., 2016), but we note Chu et al. (2016) highlight changes at only a few outlets and under the extreme scenario of $k = 1.11$ describing an over-pressured system. Because $\rho_w/\rho_i = 1.09$, setting $k = 1.09$ reduces Eq. 2 to $h = z_s$, and is equivalent to an over-pressured system with surface routing of the water. In a limited examination comparing our results with $k \in [0.8, 0.9, 1.0]$, we did not detect basins with large changes in upstream area. In addition, all time series graphics show the mean RCM discharge for $k = 1.0$, but the uncertainty among all three $k$ values (not shown) is small enough that it is difficult to distinguish the three separate uncertainty bands - the difference between RCMs or between RCMs and observations is much larger than uncertainty from the $k$ parameter.

The above issues are specific to ice basins. Land basin outlets do not change location, and the range of upstream runoff from different $k$ simulations to a land outlet provides one metric of uncertainty introduced by the $k$ parameter. This uncertainty

among all three $k$ values is small at ice margin outlets. It is even smaller at land outlets which act as spatial aggregators and increase the signal-to-noise ratio.

Below, we discuss the known uncertainties, ranging from least to most uncertain.

The basins presented here are static approximations based on 100 m DEM of a dynamic system. Land basin boundaries are likely to be more precise and accurate than ice basins, because the land surface is better resolved, has larger surface slopes, has negligible sub-surface flow, and is less dynamic than the ice surface. Even if basins and outlets seem visually correct from the 100 m product, the basin outline still has uncertainty on the order of hundreds of meters and will therefore include many minor errors and non-physical properties, such as drainage basin boundaries bisecting lakes. However, all artefacts we did find are significantly smaller than the 1 km$^2$ grid of the RCM inputs. We do not show but note that when doing the same work with the 150 m BedMachine land surface DEM, some basins change their outlet locations significantly - draining on the opposite side of a spit or isthmus and into a different fjord than the streams do when observed in satellite imagery. We have not observed these errors in streams and basins derived from the 100 m ArcticDEM in a visual comparison with Google Earth, although they may still exist.

Moving from land basins to subglacial ice basins, the uncertainty increases because subglacial routing is highly dynamic on timescales from minutes to seasons (e.g. Werder et al. (2013)). This dynamic system may introduce large spatial changes in outflow location (water or basin "piracy", Ahlstrøm et al. (2002); Lindbäck et al. (2015); Chu et al. (2016)), but Stevens et al. (2018) suggests basins switching outlet locations may not be as common as earlier work suggests, and our sensitivity analysis suggests that near the margin where the majority of runoff occurs, outlet location often changes by less than 10 km under different routing assumptions (Fig. 2). The largest (> 100 km) changes in outlet location in Fig. 2 occur when the continental or ice flow divides move, and one or two of the $k$ scenario(s) drain cells to an entirely different coast or sector of the ice sheet.

The regions near the domain edges - both the land coast and the ice margin - are covered by many small basins, and in this work basins < 1 km$^2$ are absorbed into their largest neighbor (see Methods section). By definition these basins are now hydraulically incorrect. An example can be seen in the Zackenberg basin (Fig. 16a, southwest corner of the basin), where one small basin on the southern side of a hydraulic divide was absorbed into the large Zackenberg basin that should be defined by and limited to the northern side of the mountain range.

Near the ice margin quality issues exist. At the margin, many of the small basins (absorbed or not) may be incorrect because the bed uncertainty is larger relative to the ice thickness, and therefore uncertainty has a larger influence on routing. Minor mask mis-alignments may cause hydraulic jumps (waterfalls) at the margin, or sinks that then need to be filled by the algorithm, and may overflow (i.e. the stream continues onward) somewhere at the sink edge different from the location of the real stream. The solution for individual outlets is to visually examine modeled outlet location, nearby streams in satellite imagery, and the area of upstream catchments, as we did for the Leverett Glacier outlet (Sect 4.2.3). Alternatively, selecting several outlets in an area will likely include the nearby "correct" outlet. This can be automated and an effective method to aggregate all the micro-ice basins that occur at the domain edge is to select the downstream land basin associated with one ice outlet, and then all upstream ice outlets for that land basin.

### 4.3.3 RCM uncertainty

In addition to the basin delineation issues discussed above, the runoff product from the RCMs also introduces uncertainty into
the product generated here. The RCM input products do not provide formal time- or space-varying error estimates, but of course
do contain errors because they represent a simplified and discretised reality. RCM uncertainty is shown here with a value of
±15 %. The MAR uncertainty comes from an evaluation by the Greenland SMB Model Intercomparison Project (GrSMBMIP;
Fettweis et al. (2020)) that examined the uncertainty of modelled SMB for 95 % of the 10767 in-situ measurements over the
main ice sheet. The mean bias between the model and the measurements was 15 % with a maximum of 1000 mmWE yr$^{-1}$.
GrSMBMIP uses integrated values over several months of SMB, suggesting larger uncertainty of modeled runoff at the daily
time scale. The RACMO uncertainty comes from an estimated average 5% runoff bias in RACMO2.3p2 compared to annual
cumulative discharge from the Watson River (Noël et al., 2019). The bias increases to a maximum of 20 % for extreme runoff
years (e.g. 2010 and 2012), so here we select 15 %, a value between the reported 5 % and the maximum 20 % that matches
the MAR uncertainty. We display ±15 % uncertainty in the graphics here and suggest this is a minimum value for daily runoff
data.

The 15 % RCM uncertainty is represented graphically in the time series plots when comparing to each of the observations.
It is not shown in the scatter plots because the log-log scaling and many points makes it difficult to display. In the time series
plots, we show the mean value from the $k = 1.0$ scenario, and note that discharge from the other two $k$ scenarios covered
approximately the same range.

### 4.3.4 Observational uncertainty

When comparing against observations, additional uncertainty is introduced because the stage-discharge relationship is neither
completely precise or accurate. We use published observation uncertainty when it exists. Only two observational data sets come
with uncertainty: Watson River and Qaanaaq. Similar to the RCM uncertainty, they are displayed in the time series but not in
the scatter plots.

### 4.3.5 Mitigating uncertainties

Traditional uncertainty propagation is further complicated because it is not clear to what extent the three uncertainties (obser-
vational, RCM, and routing model) should be treated as independent from each other - all three uncertainties are likely to show
some correlation with elevation, slope, air temperature, or other shared physical processes.

Many of the uncertainties discussed here can be mitigated by increasing the signal to noise ratio of the product. Because we
provide a high spatial and temporal resolution product, this is equivalent to many signals, each of which has some uncertainty
(noise). Averaging results spatially or temporally, if possible for a downstream use of this product, will increase the signal to
noise ratio and reduce uncertainty.

For example, because we provide basins for the entire ice sheet, total discharge is not subject to basin uncertainty. Any error
in the delineation of one basin must necessarily be corrected by the inclusion (if underestimate) or exclusion (if overestimate) of

460 a neighboring basin, although neighboring basins may introduce their own errors. Therefore, summing basins reduces the error introduced by basin outline uncertainty, and should be done if a downstream product does not need an estimate of discharge from a single outlet. This feature is built-in to coastal outlet discharge which is not as sensitive to our routing algorithm as ice margin outlet discharge because most coast outlets include a range of upstream ice margin outlets (e.g. Fig. 7 v. 9). Conversely, at the ice margin, outlet location and discharge volume is more uncertain than at the land coast. However, most runoff is

465 generated near the ice margin and as runoff approaches the margin, there are fewer opportunities to change outlet location (Fig. 2).

  Our coverage algorithm (Sect 3.3) only fills in glaciated regions that have at least some RCM coverage. When working with basins that have glaciated areas and no RCM coverage as in the case for all four of the GEM outlets near Nuuk, discharge could be approximated by estimating discharge from the nearest covered glaciated area with a similar climatic environment.

Temporally, errors introduced by this study's assumption of instantaneous discharge can be reduced by summing or averaging discharge over larger time periods, or applying a lag function to the time series as done here and in van As et al. (2017). Although a given volume of water may remain in storage long term, if one assumes that storage is roughly steady state, then long-term storage shown by, for example, dye trace studies, can be ignored - the volume with the dye may be stored, but a similar volume should be discharged in its place.

### 4.3.6 Quality control

The scale of the data are such that manual editing to remove artifacts is time and cost prohibitive. Here we provide one example of incorrect metadata. The elevation of each outlet is included as metadata by looking up the bed elevation in the BedMachine data set at the location of each outlet. Errors in BedMachine or in the outlet location (defined by the GIMP ocean mask) introduce errors in outlet elevation.

A large basin in NW Greenland has metadata outlet elevation > 0 (gray in Fig. 1) but appears to be marine terminating when viewed in satellite imagery. Elsewhere the land vs. marine terminating color coding in Fig. 1 appears to be mostly correct, but this view only provides information about the sign of the elevation, not the magnitude (i.e. if the reported depth is correct). Ice outlets can occur above, at, or below 0 m. It is easier to validate the land terminating basins, which should in theory all have an outlet elevation of 0 m. That is not the case (Fig. 18). It is possible for land outlets to be correctly assigned an elevation

> 0 m, if a land basin outlet occurs at a waterfall off a cliff (as might occur the edges of Petermann fjord) or due to DEM discretisation of steep cells. However, most of the land outlets at elevations other than 0 are likely due to mask misalignment placing a section of coastline in a fjord (negative land elevation) or inland (positive land elevation). The bulk of land discharge (75 %) occurs within 0 ±10 m elevation, and 90 % within 0 ±30 m elevation (Fig. 18).

### 4.4 Other sources of freshwater

The liquid water discharge product provided here is only one source of freshwater that leaves the ice sheet and affects fjords and coastal seas. The other primary freshwater source is iceberg calving and submarine melt at the ice/ocean boundary of marine terminating glaciers. A companion to the liquid water discharge product introduced here is provided by Mankoff et al.

(2019, 2020), which estimates solid ice volume flow rates across gates near marine terminating glaciers. That downstream ice enters fjords as either calving icebergs or liquid water from submarine melting.

Both this product and Mankoff et al. (2020) provide liquid or solid freshwater volume flow rates at outlets (this product) or grounding lines (Mankoff et al., 2020), but actual freshwater discharge into a fjord occurs at a more complicated range of locations. Solid ice melts throughout the fjord and beyond (e.g. Enderlin et al. (2016); Moon et al. (2017)), and the freshwater discharge presented here may enter at the reported depth (Sect. 4.3.6), but rapidly rises up the ice front and eventually flows into the fjord at some isopycnal (see Mankoff et al. (2016)). The eventual downstream location of the fresh water is not addressed in this work.

Freshwater inputs directly to the water surface are also not included in this product. The flux (per square meter) to the water surface should be similar to the flux to the non-ice-covered land surface - assuming the orographic effects on precipitation produce similar fluxes to the near-land water surface.

Finally, basal melt from 1) geothermal heating (e.g. Fahnestock et al. (2001)) 2) frictional heating (e.g. Echelmeyer and Harrison (1990)) and 3) viscous heat dissipation from runoff (c.f. Mankoff and Tulaczyk (2017)) contributes additional discharge (see for example Jóhannesson et al. (2020)) to the surface melt. Geothermal and frictional heating are approximately in steady state and contribute freshwater throughout the winter months.

## 4.5 Summary

Of the 20 comparisons between the two RCMs and the 10 observations,

- In general this product shows good agreement between observations and the modeled discharge from the RCM runoff routed to the outlets, when comparing across multiple basins, especially when ignoring small basins with small glaciers that are not included in the RCMs (Fig. 3). The agreement is not as good when estimating the discharge variability within individual basins. From this, the product is more appropriately used to estimate the magnitude of the discharge from any individual basin, and perhaps provide some idea of the statistical variability, but not necessarily the precise amount of discharge for any specific day, because routing delays are neglected.

- The majority of the 20 comparisons have the 5 to 95 % prediction interval between scales of 1:5 and 5:1. From this, the model results match observations within plus-or-minus a factor of five, or half an order-of-magnitude. Put differently, the daily RCM values for single or few basins have an uncertainty of +500 % or -80 %.

- The uncertainty of +500%/-80% is for "raw" data: daily discharge for one or few basins with a simple temporal smooth. When averaging spatially or temporally over larger areas or longer times, uncertainty decreases (Sect. 4.3). For example, when moving from daily data (Fig. 3) to annual sums (Fig. 4), the uncertainty is reduced to +100%/-50%.

- The two RCMs agree best with each other for the three observations dominated by large ice domains (Watson River (Sect. 4.2.2 & Fig. 7), Leverett Glacier (Sect. 4.2.3 & Fig. 9) which is a subset of the Watson River basin, and Kiattuut Sermiat (Sect. 4.2.4 & Fig. 10)). RCMs agree best with observations for ice-dominated basins with well-resolved bed

topography in BedMachine (i.e. correct basins modeled in this work) - here only Leverett Glacier (Sect. 4.2.3 & Fig. 9) meets this criterion.

– Runoff errors increase with low discharge (panels 'd' in Figs. 7, 9 to 17).

– For land basins, subglacial routing errors no longer exist, basins are well-defined, and errors are due to neglecting runoff delays or the RCM estimates of runoff.

– For ice basins, errors are dominated by basin uncertainty. Errors between similar-sized and neighboring basins are likely to offset and may even cancel each other. Even so, a conservative treatment might consider errors between basins as random and reduce by the sum of the squares when summing discharge from multiple similar-sized and neighboring basins.

## 5   Product description

These data contain a static map of Greenland's hydrological outlets, basins, and streams and a times-series of discharge from each outlet.

The output data are provided in the following formats: The stream data are provided as a GeoPackage standard GIS product and a metadata CSV that includes the stream type (start or intermediate segment), network, stream along-flow length, stream straight length, sinuosity, source elevation, outlet elevation, and a variety of stream indices such as the Strahler, Horton, Shreve,

Hack, and other parameters (Jasiewicz and Metz, 2011). We note that the subglacial streams are unvalidated with respect to actual subglacial conduits, and they should be used with caution. The outlet data are also provided as a GeoPackage and CSV, each of which include the outlet ID (linked to the basin ID), the longitude, latitude, EPSG:3413 x and y, and the outlet elevation. The outlet elevation is the BedMachine bed elevation at the outlet location, and users should be aware of quality issues identified in Sect. 4.3.6. The ice outlet metadata includes the ID, lon, lat, x, and y of the downstream land outlet, if one

exists. The basin product GeoPackage includes the geospatial region that defines the basin. The metadata CSV includes the basin ID (linked to the outlet ID), and the area of each basin. The time-series discharge product is provided as four NetCDF files per year, one for each domain (ice margin, land coast) and one for each RCM (MAR and RACMO). The NetCDF files contain an unlimited time dimension, usually containing 365 or 366 days, much of the same metadata as the outlets CSV file, including the outlet (a.k.a station) ID, the latitude, longitude, and elevation of the outlet, and a runoff variable with dimensions

(station, time) and units $m^3$ $s^{-1}$.

### 5.1   Database access software

The data can be accessed with custom code from the raw data files. However, to support downstream users we provide a tool to access the outlets, basins, and discharge for any region of interest (ROI). The ROI can be a point, a list describing a polygon, or a file, with units in `longitude, latitude` (EPSG:4326) or meters (EPSG:3413). If the ROI includes any land basins,

an option can be set to include all upstream ice basins and outlets, if they exist. The script can be called from the command line

(CLI) and returns CSV formatted tables, or within Python and returns standard Python data structures (from the `GeoPandas` or `xarray` package).

For example, to query for discharge at one point (50.5 °W, 67.2 °N), the following command is issued:

```
python ./discharge.py --base ./freshwater --roi=-50.5,67.2 --discharge,
```

where `discharge.py` is the provided script, `./freshwater` is the folder containing the downloaded data, and `--discharge` tells the program to return RCM discharge (as opposed to `--outlets` which would return basin and outlet information). The program documentation and usage examples are available at http://github.com/mankoff/freshwater (Mankoff, 2020b).

Because the `--upstream` option is not set, the `--discharge` option is set, and the point is over land, the results of this command are a time series for the MAR and RACMO land discharge for the basin containing this point. A small subset (the

565 first 10 days of June 2012) are shown as an example:

| time | $MAR_{land}$ | $RACMO_{land}$ |
|---|---|---|
| 2012-06-01 | 0.043025 | 0.382903 |
| 2012-06-02 | 5.5e-05 | 0.095672 |
| 2012-06-03 | 5e-05 | 0.009784 |
| 2012-06-04 | 9e-06 | -0.007501 |
| 2012-06-05 | 0.008212 | 0.007498 |
| 2012-06-06 | 28.601947 | 0.607345 |
| 2012-06-07 | 0.333926 | 0.05691 |
| 2012-06-08 | 0.489437 | 0.204384 |
| 2012-06-09 | 0.038816 | 0.167325 |
| 2012-06-10 | 5.1e-05 | 0.011415 |

If the `upstream` option is set, two additional columns are added: One for each of the two RCM ice domains. A maximum of six columns may be returned: 2 RCM times (1 land + 1 ice + 1 upstream ice domain), because results are summed across all outlets within each domain when the script is called from the command line (summing is not done when the script is accessed

from within Python).

If the `--outlets` option is set instead of the `--discharge` option, then results are a table of outlets. For example, moving 10 ° east over the ice,

```
python ./discharge.py --base ./freshwater --roi=-40.5,67.2 --outlets
```

results in

| index | id | lon | lat | x | y | elev | domain | upstream | $coast_{id}$ | ... |
|---|---|---|---|---|---|---|---|---|---|---|
| 0 | 118180 | -38.071 | 66.33 | 313650 | -2580750 | -78 | land | False | -1 | ... |
| 1 | 67133 | -38.11 | 66.333 | 311850 | -2580650 | -58 | ice | False | 118180 | ... |

If the script is accessed from within Python, then the `discharge` option returns an `xarray Dataset` of discharge, without aggregating by outlet, and the `outlets` option returns a `GeoPandas GeoDataFrame`, and includes the geospatial location of all outlets and outline of all basins, and can be saved to GIS-standard file formats for further analysis.

## 6   Conclusions

We provide a 100 m spatial resolution data set of streams, outlets, and basins, and a 1 day temporal resolution data set of discharge through those outlets for the entire ice-sheet area from 1958 through 2019. Access to this database is made simple for non-specialists with a Python script. Comparing the two RCM-derived discharge products to 10 gauged streams shows the uncertainty is approximately plus-or-minus a factor of five, or half an order-of-magnitude, or +500%/-80%, when comparing daily discharge for single or few basins.

Because of the high spatial (individual basins) and temporal (daily) resolution, larger uncertainty exists than when working over larger areas or time-steps. These larger areas and times can be achieved through spatial/temporal aggregating or by implementing a lag function.

This liquid freshwater volumetric flow rate product is complemented by a solid ice discharge product (Mankoff et al., 2020). Combined, these provide an estimate of the majority of freshwater (total solid ice and liquid) flow rates from the Greenland 590   ice sheet into fjords and coastal seas, at high temporal resolution and process-level spatial resolution (i.e. glacier terminus for solid ice discharge, stream for liquid discharge).

This estimate of freshwater volume flow rate into Greenland fjords aims to support further studies of the impact of freshwater on ocean physical, chemical, and biological properties; fjord nutrient, sediment, and ecosystems; and larger societal impacts of freshwater on the fjord and surrounding environments.

## 595   7   Code and data availability

The data from this work is available at doi:10.22008/promice/freshwater (Mankoff, 2020a).

The code and a website for post-publication updates is available at https://github.com/mankoff/freshwater (Mankoff, 2020b) where we document changes to this work and use the GitHub Issues feature to collect suggested improvements, document those improvements as they are implemented, document problems that made it through review, and mention related. This version of 600   the document is generated with git commit version 831ce52 .

# 8 Figures

## 8.1 Overview

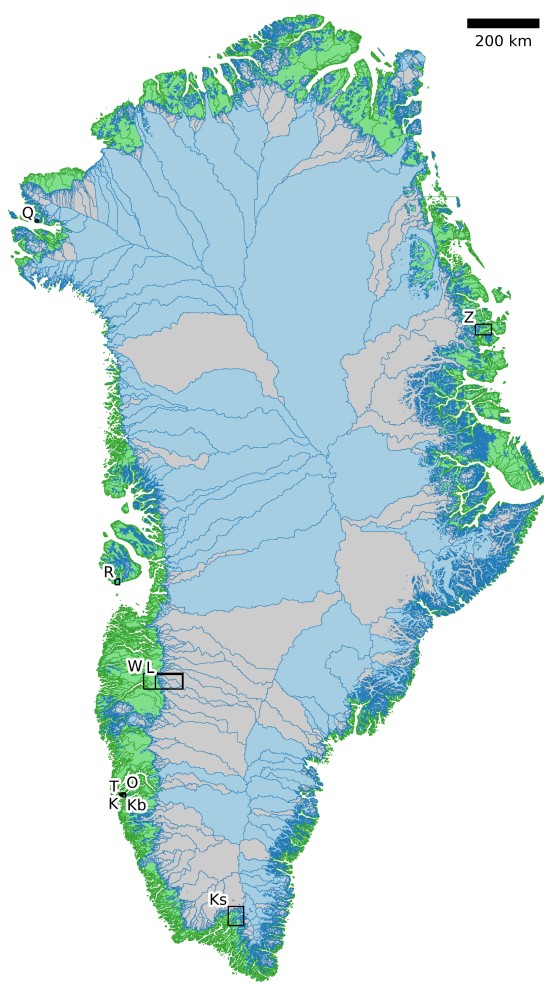

**Figure 1.** Map of Greenland showing all basins and the location of 10 gauged streams used for comparison. Land basins shown in green. Ice basins in blue when outlet elevation < 0, and gray when outlet elevation >= 0 (outlet error elevation is discussed in Sect. 4.3.6). Black boxes and labels mark location of stream gauge observation locations (see Table 1).

## 8.2 Basin changes with changing k

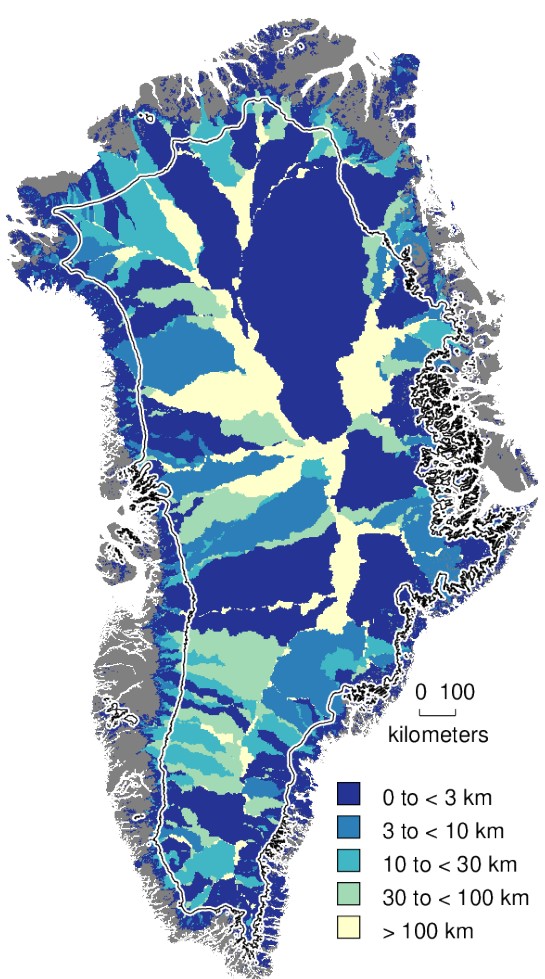

**Figure 2.** Map of Greenland showing maximum of all possible distances among outlet cell locations for all upstream cells, based on three effective basal pressure regimes ($k \in [0.8, 0.9, 1.0]$, Eq 2). Contour line shows 1500 m elevation contour - most runoff occurs below this elevation.

## 8.3 Bulk observation v. RCM scatter plots

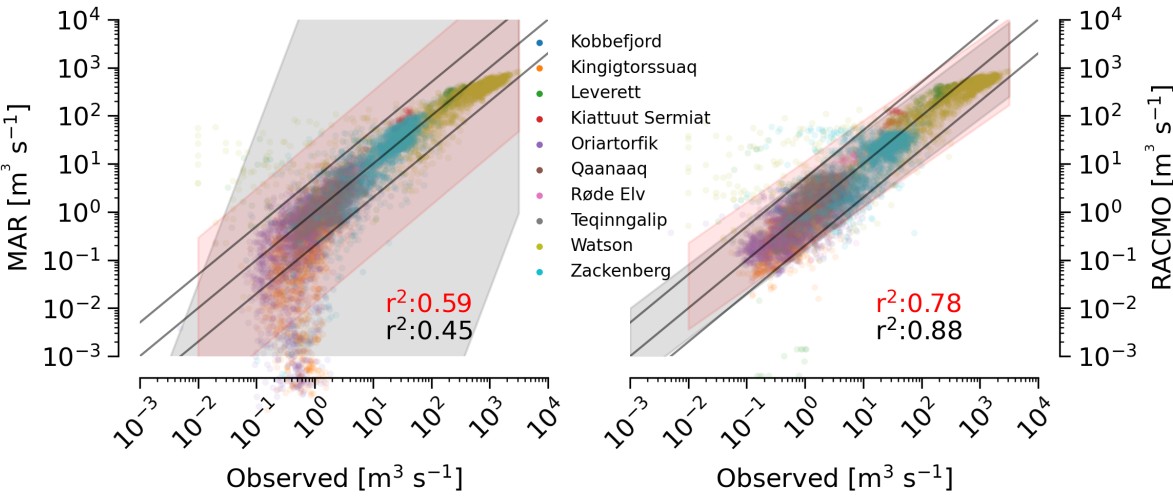

**Figure 3.** Daily runoff vs. observations for 10 outlets and a total of 15778 days. Solid lines show 1:1 (center), 1:5 (upper), and 5:1 (lower). Grey band shows 5 to 95 % prediction interval. Red band shows 5 to 95 % prediction interval when removing the GEM stations near Nuuk (Table 1) that have small glaciers not included in the RCMs (5341 days remain).

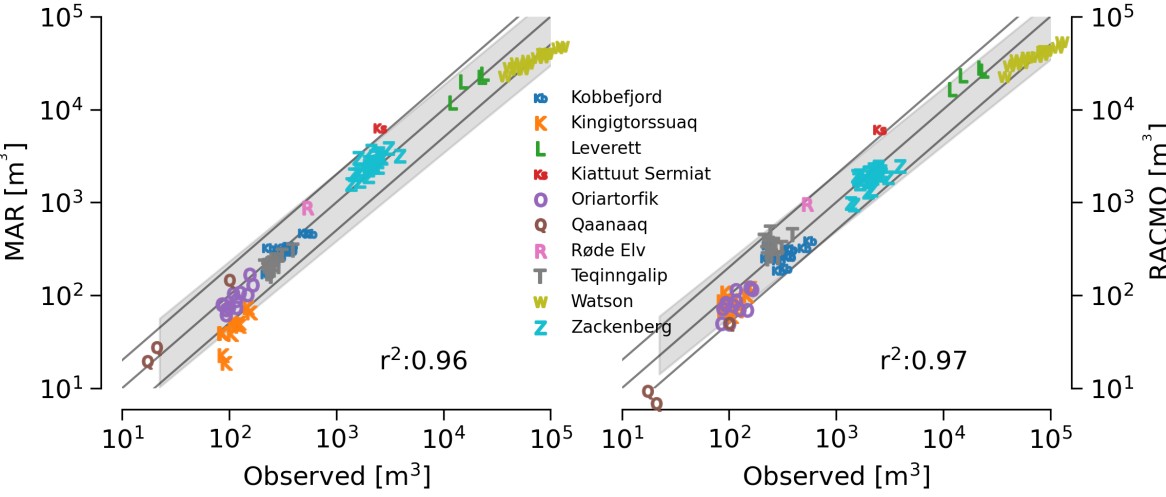

**Figure 4.** Similar to Figure 3, except here showing annual sum of observed runoff - all days within each year when observations exist are summed. Days without observation are excluded from this comparison. Solid lines show 1:1 (center), 1:2 (upper), and 2:1 (lower). Grey band shows 5 to 95 % prediction interval.

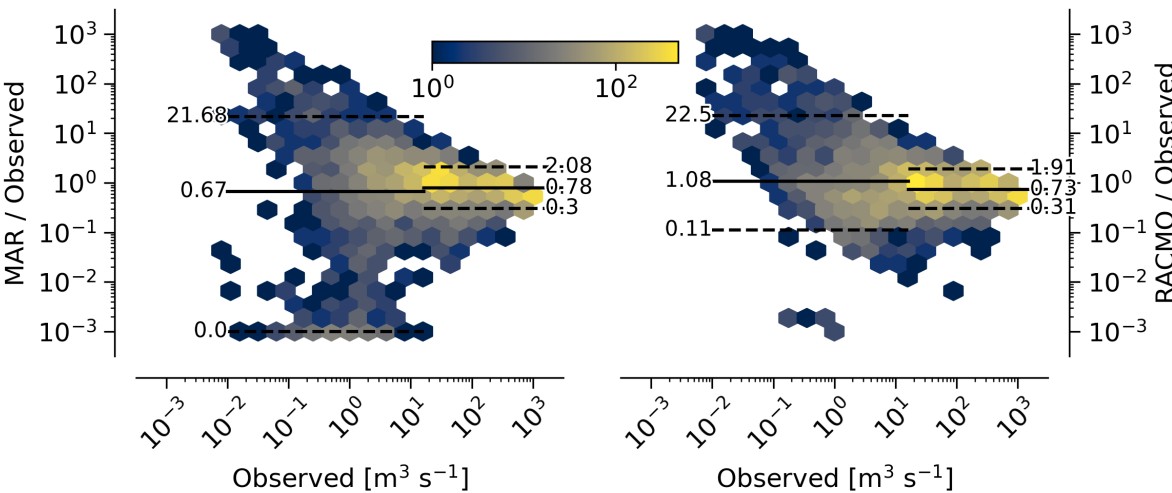

**Figure 5.** Observation vs. ratio of RCM to observations for MAR (left) and RACMO (right), discussed in Sect. 4.2.1. Number of samples at a location is represented by color. Horizontal solid line shows mean, dashed lines 5 to 95 % quantile range, and horizontal split denotes the bottom 1/3 and top 2/3rds quantiles of observed discharge. The four near-Nuuk GEM basins which have glaciers not included in the RCM domain are excluded.

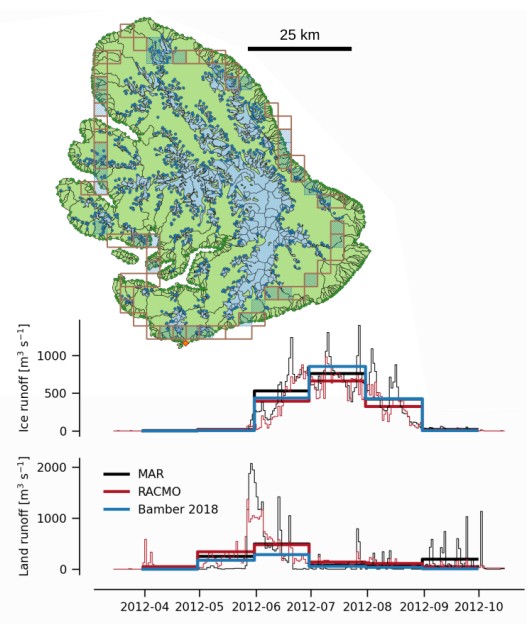

**Figure 6.** Disko Island comparison between this product and Bamber et al. (2018). Light green are land basins with dark green outlet dots. Light blue are ice basins with dark blue outlet dots. Brown and hatched blue 5 km$^2$ cells are the land and ice runoff locations, respectively, from Bamber et al. (2018). Bottom graphs show ice (upper) and land (lower) runoff for the 2012 runoff calendar year.

## 8.6 Watson River

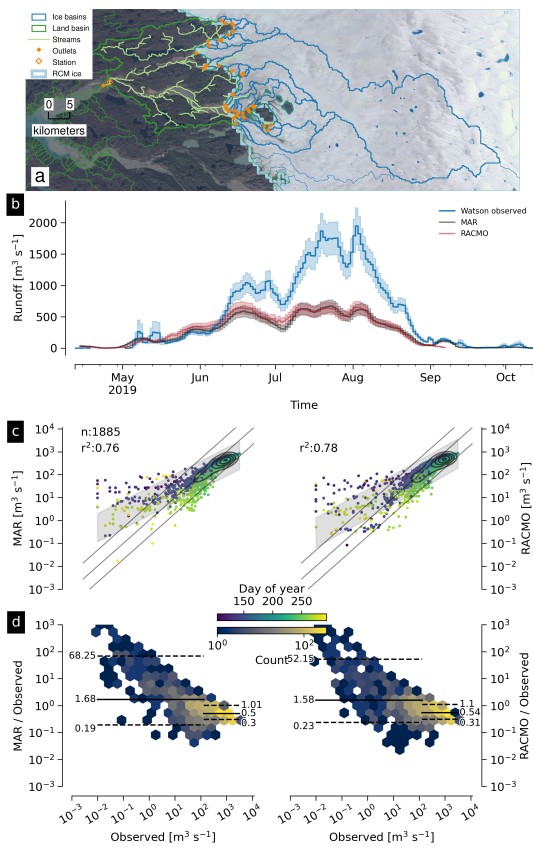

**Figure 7.** Graphical summary of Watson River outlet, basin, and discharge (W in Fig. 1). See Sect. 3.4 for general overview of graphical elements, and Sect. 4.2.2 for discussion of the Watson River basin. Basemap from Howat et al. (2014); Howat (2017a).

## 8.7 Watson Adjustments

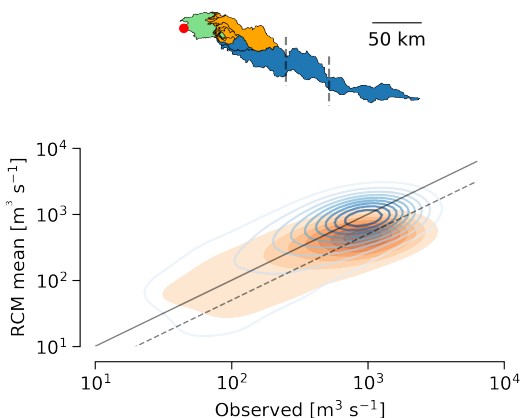

**Figure 8.** Watson River and manually adjusted basin area. Top panel: map view showing land and ice basin from this work (green and orange, respectively, same as region shown in 7, and two additional basins to the south in blue. Vertical dashed lines denote approximate location of 1500 m and 1850 m elevation. Bottom panel: Kernel density estimate (concentration of points) comparing observed vs. average of RACMO and MAR RCM runoff for the default land and ice basin (orange; filled) and with the additional southern basins (blue; lines). Solid and dashed lines are 1:1 and 2:1 (respectively) observed-to-RCM ratios.

## 8.8 Leverett Glacier

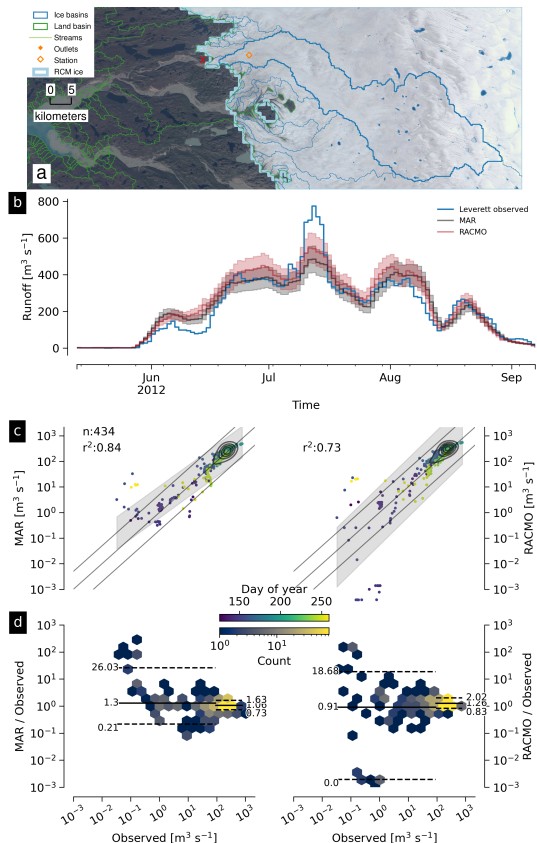

**Figure 9.** Graphical summary of Leverett Glacier outlet, basin, and discharge (L in Fig. 1). Red X in panel A marks actual observation location, but adjusted here to orange diamond within the ice basin. See Sect. 3.4 for general overview of graphical elements, and Sect. 4.2.3 for discussion of the Leverett Glacier basin. Basemap from Howat et al. (2014); Howat (2017a).

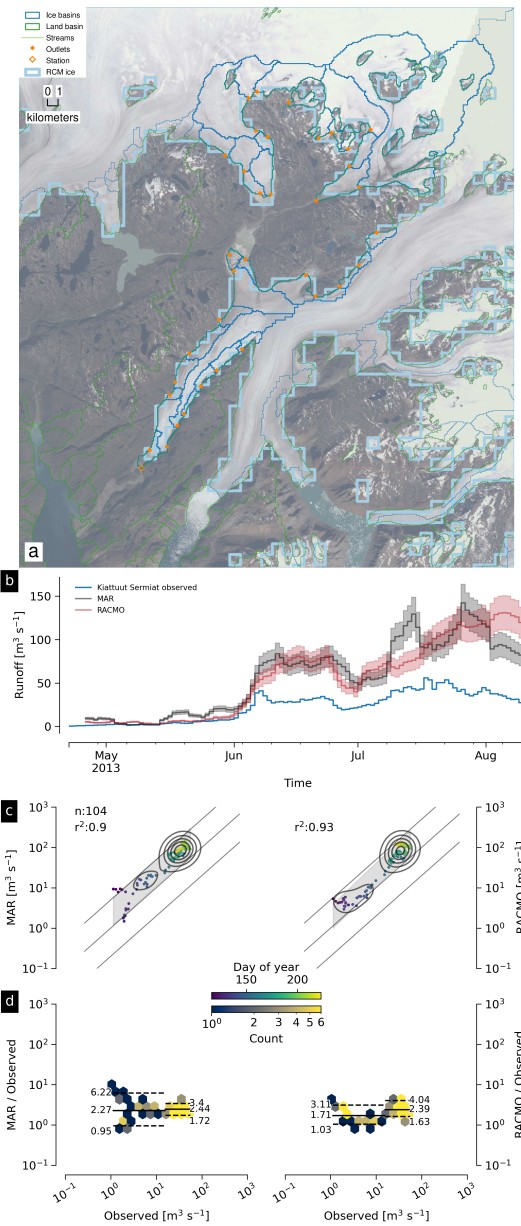

**Figure 10.** Graphical summary of Kiattuut Sermiat outlet, basin, and discharge (Ks in Fig. 1). See Sect. 3.4 for general overview of graphical elements, and Sect. 4.2.4 for discussion of the Kiattuut Sermiat basin. Basemap from Howat et al. (2014); Howat (2017a).

## 8.10 Kingigtorssuaq

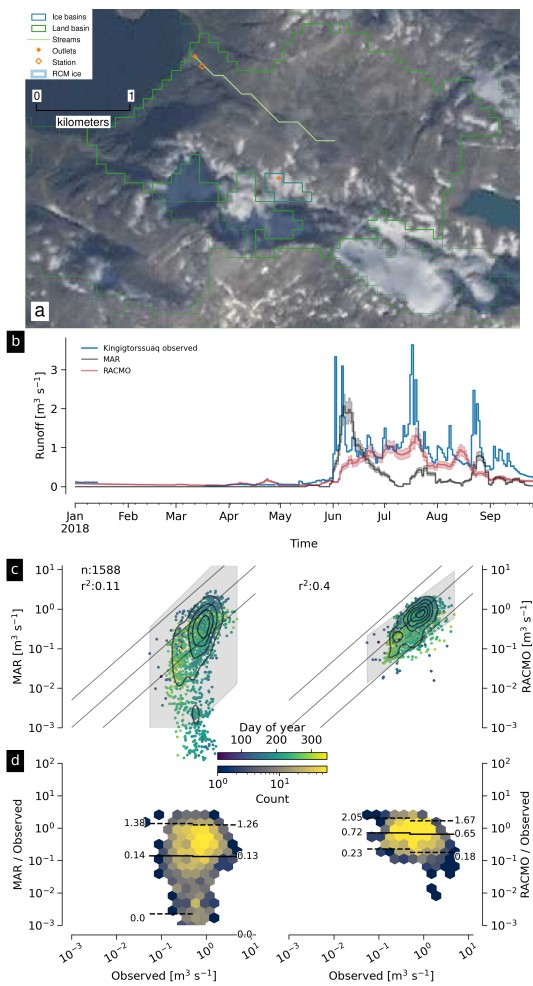

**Figure 11.** Graphical summary of Kingigtorssuaq outlet, basin, and discharge (K in Fig. 1). See Sect. 3.4 for general overview of graphical elements, and Sect. 4.2.5 for discussion of the Kingigtorssuaq basin. Basemap from Howat et al. (2014); Howat (2017a).

## 8.11 Oriartorfik

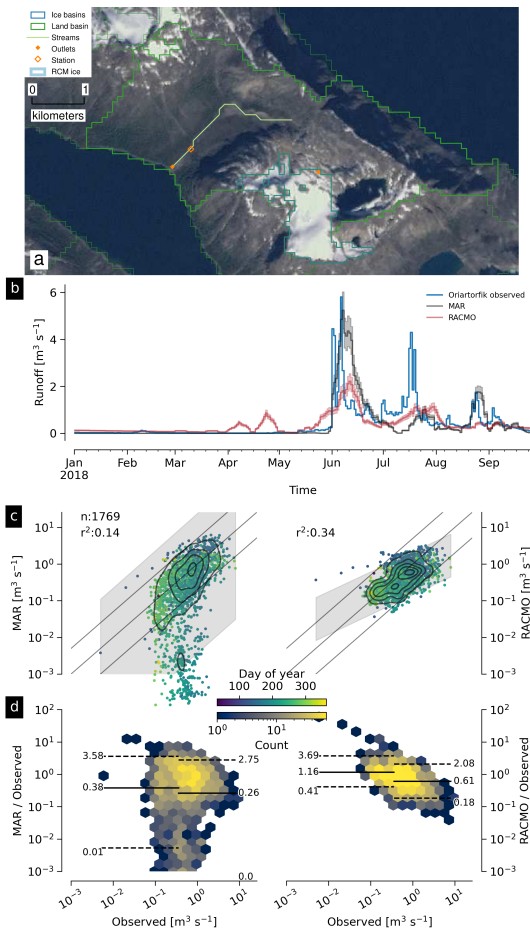

**Figure 12.** Graphical summary of Oriartorfik outlet, basin, and discharge (O in Fig. 1). See Sect. 3.4 for general overview of graphical elements, and Sect. 4.2.5 for discussion of the Oriartorfik basin. Basemap from Howat et al. (2014); Howat (2017a).

## 8.12 Teqinngalip

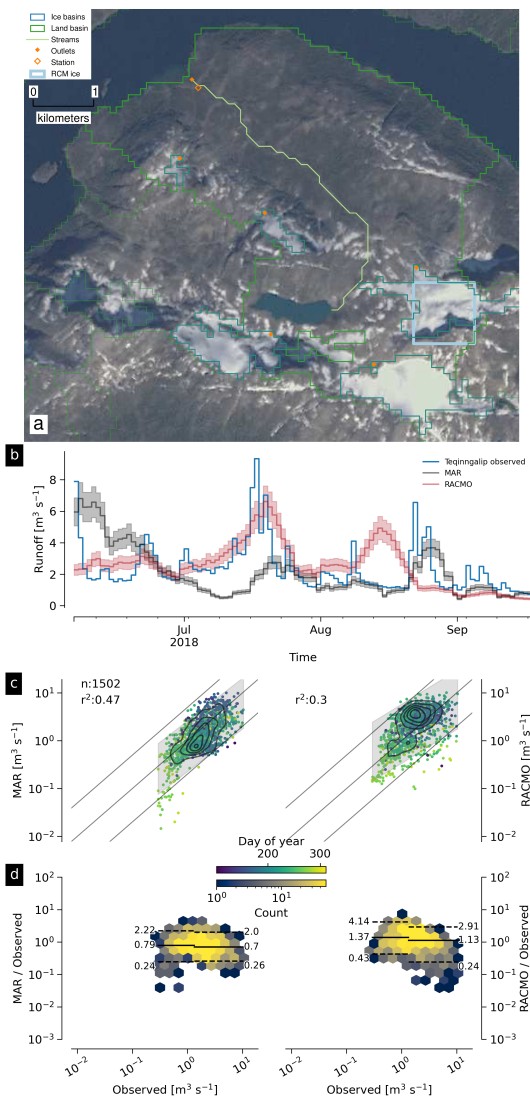

**Figure 13.** Graphical summary of Teqinngalip outlet, basin, and discharge (T in Fig. 1). See Sect. 3.4 for general overview of graphical elements, and Sect. 4.2.5 for discussion of the Teqinngalip basin. Basemap from Howat et al. (2014); Howat (2017a).

## 8.13 Kobbefjord

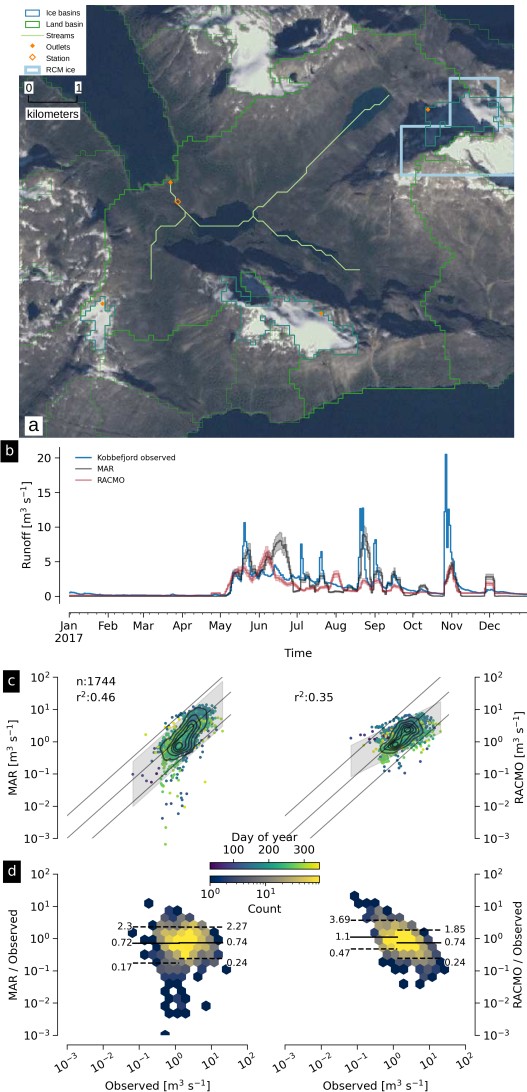

**Figure 14.** Graphical summary of Kobbefjord outlet, basin, and discharge (Kb in Fig. 1). See Sect. 3.4 for general overview of graphical elements, and Sect. 4.2.5 for discussion of the Kobbefjord basin. Basemap from Howat et al. (2014); Howat (2017a).

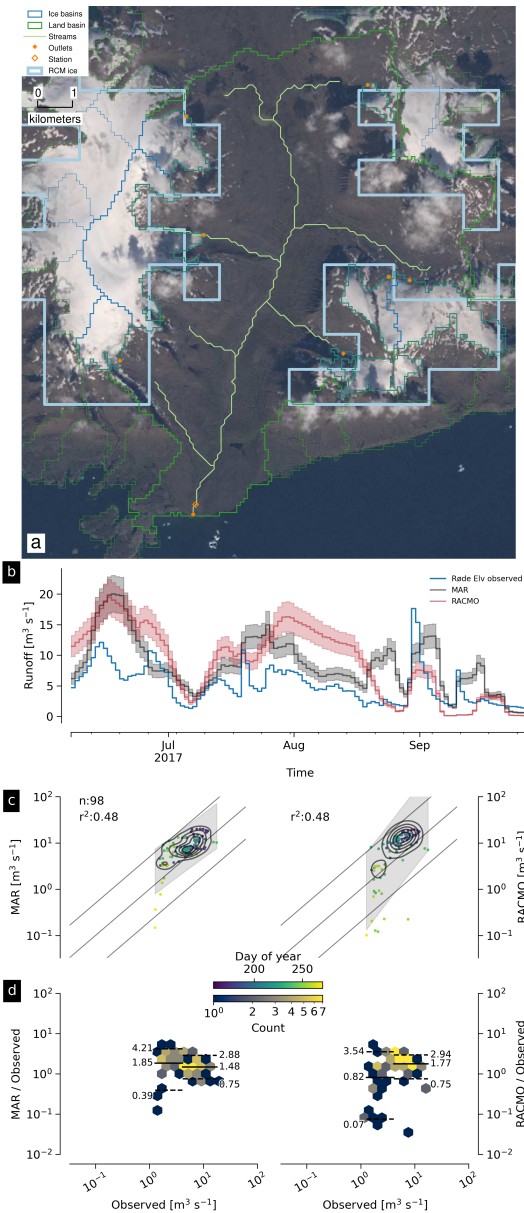

**Figure 15.** Graphical summary of Røde Elv outlet, basin, and discharge (R in Fig. 1). See Sect. 3.4 for general overview of graphical elements, and Sect. 4.2.6 for discussion of the Røde Elv basin. Basemap from Howat et al. (2014); Howat (2017a).

## 8.15 Zackenberg

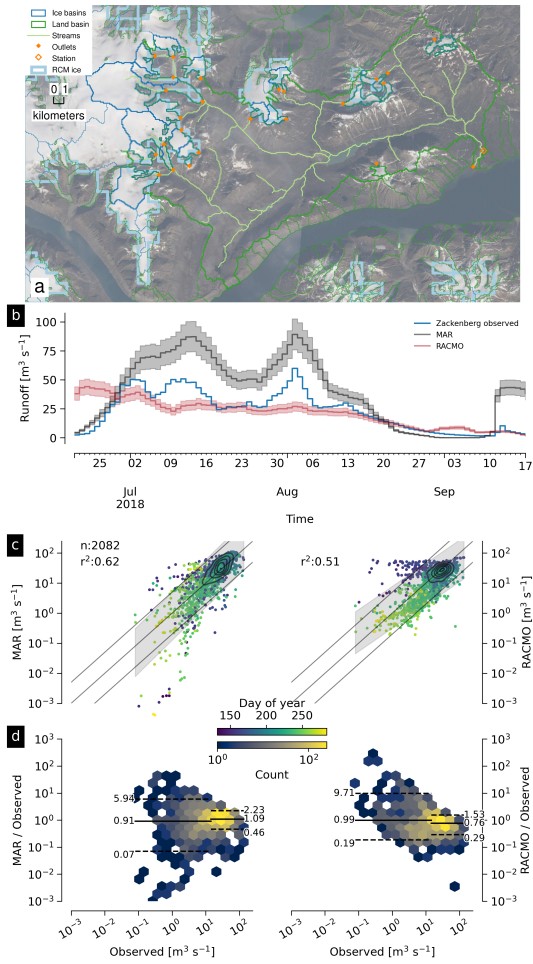

**Figure 16.** Graphical summary of Zackenberg outlet, basin, and discharge (Z in Fig. 1). See Sect. 3.4 for general overview of graphical elements, and Sect. 4.2.6 for discussion of the Zackenberg basin. Basemap from Howat et al. (2014); Howat (2017a).

## 8.16 Qaanaaq

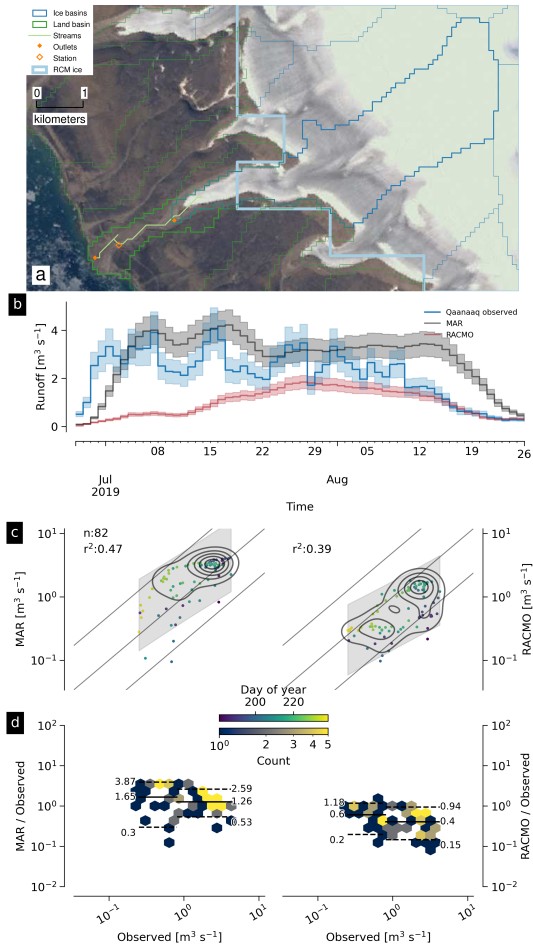

**Figure 17.** Graphical summary of Qaanaaq outlet, basin, and discharge (Q in Fig. 1). See Sect. 3.4 for general overview of graphical elements, and Sect. 4.2.6 for discussion of the Qaanaaq basin. Basemap from Howat et al. (2014); Howat (2017a).

## 8.17 Elevation histogram

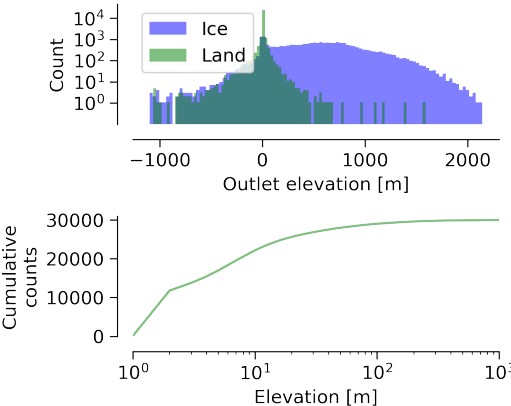

**Figure 18.** Top: Histogram of outlet elevations. Bottom: Cumulative distribution of absolute land outlet elevation. More than 75 % of land outlets occur within ±10 m, and 90 % within ±30 m.

## Appendix A: Software

This work was performed using only open-source software, primarily `GRASS GIS` (Neteler et al., 2012), CDO (Schulzweida, 2019), NCO (Zender, 2008), GDAL (GDAL/OGR contributors, 2020), and `Python` (Van Rossum and Drake Jr, 1995), in particular the `Jupyter` (Kluyver et al., 2016), `dask` (Dask Development Team, 2016; Rocklin, 2015), `pandas` (McKinney, 2010), `geopandas` (Jordahl et al., 2020), `numpy` (Oliphant, 2006), `x-array` (Hoyer and Hamman, 2017), and `Matplotlib` (Hunter, 2007) packages. The entire work was performed in `Emacs` (Stallman, 1981) using `Org Mode`

(Schulte et al., 2012) on GNU/Linux and using many GNU utilities (See Supplemental Material). The `parallel` (Tange, 2011) tool was used to speed up processing. We used `proj4` (PROJ contributors, 2018) to compute the errors in the EPSG 3413 projection. The color map for Fig. 2 comes from Brewer (2020).

All code used in this work is available in the Supplemental Online Material.

*Author contributions.* KDM produced this work - wrote the code and the text. XF and BN supplied RCM inputs. APA, WIC, DVA, and RSF

helped with discussions of methods, quality control, or writing. KK and SS supplied Qaanaaq data. KL provided GEM data.

*Competing interests.* The authors declare that they have no conflict of interest.

*Acknowledgements.* Funding was provided by the Programme for Monitoring of the Greenland Ice Sheet (PROMICE). Parts of this work were funded by the INTAROS project under the European Union's Horizon 2020 research and innovation program under grant agreement No. 727890. BN was funded by NWO VENI grant VI.Veni.192.019. DEMs were provided by the Polar Geospatial Center under NSF-OPP

awards 1043681, 1559691, and 1542736. Data from the Greenland Ecosystem Monitoring Programme (GEM) were provided by Asiaq – Greenland Survey, Nuuk, Greenland. We thank Dorthe Petersen (Asiaq) for help with basin quality control. The editor and two anonymous reviewers provided valuable feedback and helped improve this paper (Anonymous, 2020a, b).

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
