# Peer review of "Greenland liquid water discharge from 1958 through 2019"

_Earth System Science Data, 2020_

## Referee Comment (RC1) · Anonymous Referee #1 · 27 Apr 2020

**Summary and general comments**

A high-resolution product of liquid discharge from the Greenland Ice Sheet (GrIS) and the unglaciated area of Greenland is derived for the period 1979 – 2017 and provided with various static hydrological quantities (e.g. basins and outlet locations). Gridded runoff is taken from two regional climate models (RCMs) simulations (MAR and RACMO), whose output is statistically downscaled to a horizontal resolution of 1 km. Hydrological characteristics (e.g. basin delineation) are computed from surface elevation according to ArcticDEM. BedMachine surface and bed elevation data is additionally considered to assess the sensitivity of the routing network to uncertainties in surface topography and to consider subglacial ice sheet drainage.

This study addresses a very relevant topic, namely the quantification of liquid discharge from Greenland in the current climate and specifically the locations where this freshwater will enter the ocean. It closes the link between RCM simulations, which provide gridded runoff at increasingly higher horizontal resolution and the need for (high-resolution) liquid discharge locations, which are not directly provided from RCM simulations. The manuscript is well written but the structure needs some improvements in my opinion. Additionally, certain topics (particularly methods) are not explained with enough details.

**Major comments**

**1) Improve structure of manuscript**
In my opinion, the manuscript lacks a clear structure, as e.g. introduction of data and applied methods are not restricted to the data and methods sections but also appear e.g. in section 6. Furthermore, the partitioning of subtopics in results and discussion (sections 5 and 6) does not seem logical to me. I would suggest the following structure:

1. *Introduction*
2. *Input and validation data*
    2.1. *Downscaled gridded RCM data (part of current section 2)*
    2.2. *Time-invariant data (DEMs, ice/ocean masks) (part of current section 2)*
    2.3. *River discharge observations (part of current section 6.2)*
3. *Methods*
    3.1. *Masks and grid cell alignment (current section 3.1)*
    3.2. *Derivation of hydrological quantities (e.g. basins, outlet locations, etc.) (see next major comments for more details about the content of this section) (current section 3 and part of current section 6.3.1)*
4. *Product evaluation and assessment*
    4.1. *Main characteristics (current section 5)*
    4.2. *Comparison with previous similar work (current section 6.1)*
    4.3. *Validation/Comparison of product with observational river discharge (current section 6.2)*
    4.4. *Product uncertainties (current section 6.3)*
    4.5. *Remaining sources of freshwater input in fjords (current section 6.4)*
5. *Technical product description and data/code availability*
    5.1. *Product description (current section 4)*
    5.2. *Data and code availability (current section 7)*
6. *Conclusions*

**2) Missing parts in method section**
The method section should be extended – particularly the part about the derivation of the hydraulic characteristics. Specifically, I miss information about:
- How were (artificial) depressions in the DEM handled? With a filling algorithm?
- I'm confused about the applied flow direction algorithm. Was a single flow direction (SFD) or a multiple flow direction (MFD) algorithm used? And how were the basins delineated if a MFD algorithm was used (which has a dispersive character)?

Moreover, the method used for assessing the basin uncertainty (section 6.3.1) should be moved to this section. It should include a more detailed discussion of the equation used to compute the hydraulic head and how this equation is applied to derive the sensitivity experiments in the appendix (with various subglacial pressure).

**3) Sensitivity of basins delineation to uncertainties in surface elevation and partitioning of surface/subsurface runoff**

The evaluation of the Kangerlussuaq / Watson river catchment with river discharge data reveals that the accurate basin delineation is crucial. The sensitivity experiments with a different DEM for the surface and the consideration of subglacial drainage are thus extremely interesting and useful. I wonder if the uncertainty in the basin delineation, which is illustrated in the appendix, could be translated to runoff uncertainties (and be included in the runoff output product). One could for instance compute discharge at all (coastal) outlets for all sensitivity experiments and check the range in obtained runoff. This work would reveal catchments for which runoff quantifications are more (un-)certain. It's probably not necessary to include these runoff uncertainty values in the current product but it would be nice upgrade.

**Minor comments**

**Content-related (text)**

**Page 1 line 10:** "contributes an additional ~35% to the ice runoff " → confusingly stated (because the ~35% are referring to the total runoff I guess) → rephrase

**P2L26-28:** I don't understand to what "satellite basemap imagery" is referring to. To the ocean mask?

**P2L29:** Mention somewhere here that RACMO only provides runoff for the glaciate area of Greenland

**P3L5-7:** The runoff downscaling should be explained in more detail (or a reference for the procedure should be provided)

**P3L8:** Is it justified to assume that the firn layers in both simulations (MAR and RACMO) are in approximate equilibrium in 1979 (i.e. was there a spin-up performed or when did the simulations start?)

**P3L14-17:** How are (artificial) inland depression treated that would lead to erroneous inland outlets. Are such depressions apparent in the DEM? And if so, how are they removed?

**P3L15:** I'm confused by the part "multi-flow direction from eight neighbors". Does it imply that a multi-flow direction algorithm with dispersion was used? Or a D8 algorithm (because this algorithm also allows flow **from** (maximal) eight neighbors).

**P3L26-27:** I'm not sure if I understand this sentence correctly: so land pixels surrounded by ice are set to ice (but their elevation is left unchanged)?

**P4L18-20:** I don't understand this part: Is the downscaled gridded runoff data provided on an EPSG:3413 map projection (because I guess the direct output from the RCMs is on a rotated lon/lat-grid)? And the EPSG:3413 projection is based on WGS 84 (and thus an ellipsoid). But some data is provided in a coordinate system based on a sphere (earth spheroid)?

**P4L22-23:** It should be stated in this section that land quantities (e.g. basin polygons and runoff) also include the same quantity from the glaciated part (I assume). So I guess runoff from land contains both runoff from the unglaciated and the glaciated part?

**P5L8-10:** Why are the more larger land basins than ice basins? Do the land basins incorporate the ice basins?

**P5L20-22:** It should be more clearly stated in this sentence that the 4380 $m^3$ refers to runoff from a single basin.

**P5L24:** I assume the ±30 $km^3$ represent the RCM runoff uncertainty of 15% (this should be clearly stated here). And shouldn't it rather be ±60 $km^3$? And how is this value of 15% derived (is there a reference)? I think it would be useful to mention this uncertainty value already in section 2 (input data).

**P6L8-11:** This sentence does not belong here but rather in section 6.3.1. Furthermore, I find the sentence a bit hard to understand (particularly the last part) – could it be rephrased? It states that flow-path derived from the ArcticDEM generally agree better with satellite images than flow-path derived from BedMachine data, right?

**P6L17:** Could you explain the reason why the increase in spatio-temporal resolution increases the signal-to-noise ratio in more detail? And I would include a reference to section 6.3.4 here (so that the reader knows where this strategy is discussed in the manuscript).

**P6L18-26:** I would move this part to the data section (2).

**P7L33-34:** "MAR runoff slightly overestimates the GEM observations early in the year, and slightly underestimates the observation late in the year" → this is an interesting finding and probably related to storage of water in the (un-)glaciated area of the basin on intraannual time scales

**P8L3-4:** This step-like change in MAR runoff is rather strange. Are you certain that this is not an artefact (e.g. caused by an issue in MAR, the statistical downscaling procedure of runoff or the alignment of the 1 km and the 100 m masks)?

**P8L12-13:** It seems a bit arbitrary to exclude the 27th and 28th July.

**P8L27:** "slight lag between models signals and the observations." → could this time lag be related to the neglect of routing travel time?

**P8L28-29:** What is the reason for the significantly higher temporal variability in RACMO? Could this be linked to the different treatment of liquid water retention on bare ice between the RCMs?

**P9L1-5:** Why was the existing proxy data not used for further model validation (if it exists)?

**P9L12:** There is no equation 1

**P9L15:** "because large volumes of runoff usually come from large areas." → I do not understand this part of the sentence, could it be rephrased?

**P10L4-5:** What is meant by "hydraulic jumps"? I guess not the physical phenomena in hydraulics. If not, this term should be replaced to avoid ambiguity.

**P10L11:** This equation (and the corresponding text) should be moved to the method section.

**P10L17-19:** I find the transition between the previous and this part a bit strange. The part before explains how routing and basin delineation is derived when bed elevation is considered (this part should anyway be explained in the methods section in my opinion) but this section compares basin delineation based on two different **surface** topographies.

**P10L30:** Can you provide a reference for this value of 15%? Also, this value should already be mentioned in the data description (section 2).

**P11L4-5:** Replace "highlighted above" with reference to relevant section. Additionally, are you certain that the step-like changes in RCM runoff originates from the actual RCM simulation (and is not generated by the subsequent postprocessing steps (e.g. downscaling or grid cell alignment).

**P11L15:** "current limitation" → future RCM simulation will still only capture features and process of certain spatial scales. But do you think that the most crucial scale will be represented in these simulations with higher resolution?

**P12L4-5:** Can you provide a reference that supports this (net storage is approximately zero) assumption?

**P12L21-23:** This sentence should be rephrased or removed. Making a prediction about fjord precipitation from the Greenland-wide fraction of land runoff is not reasonable in my opinion.

**P13L6-7:** "perhaps due to temporal directionality" → I don't understand this part

**P13L7:** Is "version of the dataset" meant here?

**P13L20:** Again, are the stated uncertainty values correct?

**Typos, phrasing and stylistic comments**

**Page 1 line 10:** Change "over the time series" to "over time"

**P2L4-5:** I don't understand the meaning of this sentence ("Immediately upstream from…") – isn't it obvious that no submarine melting occurs upstream of the grounding line?

**P3L12-13:** "Each outlet has one upstream basin and each basin has one outlet" → I don't understand the meaning of this sentence, isn't this fact obvious?

**P3L22:** Change "100 $m^2$ pixel" to "10,000 $m^2$ pixel"

**P4L12-13:** "In the case of a small basin," → this sentence is a bit oddly stated – could it be rephrased?

**P4L30-31:** I would remove "four per year" (and optionally change "provided as annual NetCDF files" to "provided as four annual NetCDF files").

**P5L6-7:** "Runoff ice products…" → oddly stated sentence → rephrase

**P5L21:** "2012-08-06" → write date out

**P5L27:** "contributes an additional 35% to the ice runoff" → again, I find this a bit confusingly stated (I guess the 35% refer to total liquid runoff?). Maybe better: "contributes 35% to total runoff"

**P6L5:** Maybe change "and additional data products." to "and is provided with additional data."

**P6L23:** change "results to all observations that we have been able to find that are publicly accessible" to "results to all publicly accessible observations we could find"

**P7L1:** change "with high melt or runoff; Basin" to "with high runoff (and associated melt): Basin"

**P7L7:** change "include ice to the south of itself" to "include a glaciated area to the south"

**P7L23:** "and a without an ice basin does have RCM ice cells" → odd formulation → rephrase (e.g. change "without an ice basin" to "unglaciated"

**P8L11-12:** Rephrase sentence, e.g. to "The MAR relative runoff bias ranges from -20% (last day of time series) to +140% (28 July)."

**P8L25:** change "models than the observations" to "models than in the observations"

**P8L7:** change "discussed below" to "still discussed in Sect. 6.3.2."

**P8L8:** change "source uncertainty – the routing model, which exhibits in two different ways: Spatial (basin delineation) and temporal (runoff delay)" to "source of uncertainty – the routing model, which generates both spatial (basin delineation) and temporal (runoff delay) uncertainty"

**P9L10-11:** Rephrase to e.g.: "Temporal uncertainty is not systematically addressed in this work but a method to reduce it is discussed in Sect. 6.3.4."

**P10L19-20:** Change sentence to e.g. "Results from additional sensitivity experiments (with different input data and hydraulic head computations) are shown in the Appendix."

**P10L29:** rephrase "they do not precisely nor accurately capture reality" to e.g. "they represent reality discretised and simplified."

**P12L11:** change "That ice downstream" to "The downstream ice"

**P12L26:** Change "are approximately steady state" to "are approximately in steady state"

**P12L30:** Replace "GIS-wide ice sheet surface runoff" with "Greenland-wide ice sheet surface runoff". Otherwise, "GIS" is used both for "Greenland Ice Sheet" and "Geographical Information System"

**P13L2:** Replace "This work in its entirety is available" with "Output data of this work and part of the discharge observations are available"

**P13L8-9:** This sentence is a bit oddly stated. Could you rephrase it?

**P13L12-15:** This sentence is rather complicated to read and understand. Could you rephrase it?

**P13L15:** change "differences in needed" to "differences are needed"

**P13L21:** change "displaying and overall increase in both magnitude and variability" to "an overall increase in both its magnitude and variability"

**P13L22:** change "scale" to "scales"

**Figures and Tables**
**Figure 1:** Change caption to: "Overview showing ice basins (blue), land basins (green) and locations of following map figures (black)."
**Figure 2:** Change "Sec." to "Sect." in caption (also other occurrences)
**Figure 3:** What is meant with "(this)" in the caption?
**Figure 4:** Maybe the error bars should be removed from this figure to improve readability. Additionally, "(this product based on ArcticDEM basins in Fig. 5)" should be rephrased.
**Figure 5:** It's difficult to distinguish between different basins in this plot. Maybe readability could be improved by only plotting the basin's boundaries (without hatching). Could you also plot the Lindbäck et al. (2015) basin and the one you used to produce the right panel of figure 4?
**Figure 6:** A reference to this figure only appears in section 6.3.1, so its number and position should be changed accordingly. The legend is hard to read (it could be moved outside of the map area). Additionally, I would remove the sentence: "Region is zoomed in near Sermeq Kujalleq (Jakobshavn Isbræ)."
**Figure 7:** Change "Fig. 10" to "Fig. 10 and 11" and "visible is basin artefact" to "visible is a basin artefact". Is the "RCM ice" mask showing the mask from RACMO or from MAR (also in the following figures)?
**Figure 8:** Use "Fig." or "figure" consistently.
**Figure 10:** Change "Only 2017 shown" to "Only 2017 is shown"
**Figure 11:** Again, I would remove error bars to increase readability.
**Figure 13:** Change "Uncertainty only shown for total MAR runoff, not ice or land components." to "Uncertainties are only shown for MAR total runoff and not the individual land/ice components."
**Figure 15:** This plot is very hard to read. Again, I would remove the error bars. It also difficult to distinguish MAR from RACMO. Additionally, "MAR" should be removed from the y-axis labelling.

**Supplementary material**
**Figure B1:** Remove "not zoomed in". Additionally, I would always provide all necessary information in the figure caption about the comparison (experiment setting, margin/coastal outlet). References to other figures implies constant switching between figures. This also applies for the following figures.

**Review of provided dataset**

The presented dataset provides, to my knowledge, a unique and new source for high-resolution discharge data for the GrIS and the unglaciated part of Greenland for the present-day climate (1979 – 2017). The dataset seems very useful for downstream application in various field like e.g. hydrology, ecology and oceanography (particularly for fjord systems). The dataset can be accessed via the provided link, is complete and sufficiently supported with metadata and seems to be of good quality.

However, the description of certain processing steps is insufficient in my opinion and should be improved in the manuscript (see point 2 under "Major comments").

**Minor issues**
- I was not able to found units for the Qaanaaq discharge dataset ( https://promice.org/PromiceDataPortal/api/download/0f9dc69b-2e3c-43a2-a928-36fbb88d7433/version_01/meltwater_discharge/qaanaaq)
- When checking the static data (basin polygons, outlet locations and streams), I found some inland outlet locations near Kangerlussuaq. What is the reason for this?

---

## Referee Comment (RC2) · Anonymous Referee #2 · 29 Apr 2020

This study provides high-resolution datasets of Greenland hydrologic outlets, basins, and streams, and a 1979 through 2017 time series of Greenland liquid water runoff for each outlet. This is a timely and important contribution for the Greenland hydrology community and I'm happy to see the paper and the associated datasets to be published. That said, I think some important issues need to be solved before it can be considered for publishing in ESSD.

General comments:

1. The result section does not highlight the main contribution of this study very well. It includes numerous numbers of basins, outlets, streams, runoff but their importance is not well demonstrated. Furthermore, this section focuses on the total ice and land runoffs which can be easily derived from RCMs and have been well reported in previ-

ous studies. I suggest the result section should focus on what we can learn from runoff partitions in different basins, which is the new contribution of this study.

2. The discussion section is too long and not easy to follow, particularly "6.2 Validation against observations". Most parts of section 6.2 should be removed to the result section. I suggest the authors only highlight the most important implications of their datasets and shorten this section.

3. It is important to mention that moulins are not identified so stream networks are delineated to continuously flow from inland to ice edge outlets. Therefore, the stream product may not represent the actual hydrological environments where moulins are widely distributed and fragment drainage networks, such as southwest GrIS. In contrast, the stream product may reasonably predict northwest GrIS drainage pattern since no moulins form there. Moreover, the contributing area threshold should be better illustrated since it determines the extent of streams. It may be useful to state that the derived stream product aims to represent the general meltwater flow pattern rather than the actual spatial distribution of supraglacial rivers and streams.

4. The quality of the main figures should be improved. Currently they are not satisfactory for publishing. Also, the main point of each figure should be highlighted.

5. More previous similar studies should be included. In the paper, the authors only compare their results with Lewis and Smith (2009). However, at least two important similar studies, Andersen et al (2015) and Pitcher et al (2016), should be added as comparison results.

Andersen, M.L., Stenseng, L., Skourup, H., Colgan, W., Khan, S.A., Kristensen, S.S., Andersen, S.B., Box, J.E., Ahlstrøm, A.P., Fettweis, X., Forsberg, R. Basin-scale partitioning of Greenland ice sheet mass balance components (2007–2011). Earth Planet. Sci. Lett. 2015(0): 89-95. Pitcher, L.H., Smith, L.C., Gleason, C.J. CryoSheds: a GIS modeling framework for delineating land-ice watersheds for the Greenland Ice Sheet. GIsci. Remote Sens. 2016(6): 707-722.

Minor comments:

P1 L17, in this paragraph, I think it is necessary to say surface runoff contributes very importantly to Greenland mass balance (along with ice discharge).

P2 L4, it is not straightforward to understand "liquid runoff form surface melt, condensation, and rainfall".

P2 L22, why is 100 m ArcticDEM used to do the analysis?

P3 L5, it may be worthy to mention that weathering crust of bare ice layer can store meltwater. Citation is required for this sentence.

P3 L7, citation is required for this sentence.

P3 L10, it is not common to use the term "hydrologic head elevation".

P3 L11, it is unclear how outlets are determined.

P3 L13, it is unclear what "major streams" means, some specific channel initiation thresholds (i.e. contributing area thresholds) are used to extract streams? It may be useful to call these "major streams" as rivers.

P3 L17, why is 1 km2 used as threshold to merge small basins?

P3 L20, "When this value is negative, it indicates submarine (subglacial) discharge", this sentence is not clear.

P3 L21, this section is too long. I suggest it should be shorten or some parts can be put into SUPP.

P4 L24, see my general comment, more explanations should be provided for the stream product.

P5 L8, it is not easy to understand what these numbers mean and why they are important.

P5 L15, what does "adjusts" mean here?

P5 L21, 4380 m3 rather than 4380 m-3.

P5 L23, which basin? also report the similar value in Lewis and Smith (2009).

P5 L26, Mt. Pinatubo eruption, add a citation to support this result.

P5 L27, the land runoff is considerably large. It is useful to further illustrate its meaning.

P6 L7, "Routing with a 5 km DEM is likely to cause some basins and outlets to drain into an incorrect fjord", what is the reason for this?

Fig 2 is not easy to understand. What is the main point of this figure? Perhaps remove it to the SUPP?

Fig 4, it is not clear why runoff from the Watson River basin plus the two large basins immediately to the south performs better.

Insets are required to show the location of Fig 5 -9 in Greenland.

Fig 6 is not easy to follow. What is the meaning to change outlet locations?

Fig 7 is not easy to follow. What is the main point of this figure?

Merge Figure B1-B8 into one figure.

---

## Author Comment (AC1) · 26 Aug 2020

We are grateful for the helpful reviews and pleased that the reviewers seem to think this is useful work. We would like to warn and apologize to the reviewers that due to their helpful comments, the document has changed substantially - enough that while the LaTeX diff program did not crash, it does not produce very useful results.

This significant re-write means that some of the specific comments you provided cannot be directly addressed because some text and figures have been entirely removed. In addition, reviewing this reply may be a bit more work than if we had only done the minimum needed to address the comments. We hope you don't mind this extra work and see the value in the changes that we made. Thank you.

[Figure]

Please also note the supplement to this comment:
https://essd.copernicus.org/preprints/essd-2020-47/essd-2020-47-AC1-
supplement.pdf

**Supplement:**

**Reply to Reviewers**

Ken Mankoff *et al.*

Comments from reviewers are in normal font and differentiated from the replies that use a **bold colored font.**

**Contents**

**1  Summary**

**We are grateful for the helpful reviews and pleased that the reviewers seem to think this is useful work. We would like to warn and apologize to the reviewers that due to their helpful comments, the document has changed substantially - enough that while the LATEXdiff program did not crash, it does not produce very useful results. This significant re-write means that some of the specific comments below cannot be directly addressed because some text and figures have been entirely removed, and that a second round of review may be a bit more work than if we had only done the minimum needed to address the comments. We hope you don't mind this extra work and see the value in the changes that we made. Thank you.**

**2 Review 1**

**2.1 Summary and General Commments**

A high-resolution product of liquid discharge from the Greenland Ice Sheet (GrIS) and the unglaciated area of Greenland is derived for the period 1979 – 2017 and provided with various static hydrological quantities (e.g. basins and outlet locations). Gridded runoff is taken from two regional climate models (RCMs) simulations (MAR and RACMO), whose output is statistically downscaled to a horizontal resolution of 1 km. Hydrological characteristics (e.g. basin delineation) are computed from surface elevation according to ArcticDEM. BedMachine surface and bed elevation data is additionally considered to assess the sensitivity of the routing network to uncertainties in surface topography and to consider subglacial ice sheet drainage.

This study addresses a very relevant topic, namely the quantification of liquid discharge from Greenland in the current climate and specifically the locations where this freshwater will enter the ocean. It closes the link between RCM simulations, which provide gridded runoff at increasingly higher horizontal resolution and the need for (high-resolution) liquid discharge locations, which are not directly provided from RCM simulations. The manuscript is well written but the structure needs some improvements in my opinion. Additionally, certain topics (particularly methods) are not explained with enough details.

**We're happy to read that reviewer 1 thinks the topic is relevant, and this work "closes links" and is well written. Reviewer 1 has also provided an extremely detailed review. We respond to all comments below.**

**2.2 Major Comments**

**2.2.1 1) Improve structure of manuscript**

In my opinion, the manuscript lacks a clear structure, as e.g. introduction of data and applied methods are not restricted to the data and methods sections but also appear e.g. in section 6. Furthermore, the partitioning of subtopics in results and discussion (sections 5 and 6) does not seem logical to me. I would suggest the following structure:

```
1. Introduction
2. Input and validation data
2.1. Downscaled gridded RCM data (part of current section 2)
2.2. Time-invariant data (DEMs, ice/ocean masks) (part of current section 2)
2.3. River discharge observations (part of current section 6.2)
3. Methods
3.1. Masks and grid cell alignment (current section 3.1)
3.2. Derivation of hydrological quantities (e.g. basins, outlet locations, etc.)
     (see next major comments for more details about the content of this section)
     (current section 3 and part of current section 6.3.1)
4. Product evaluation and assessment
4.1. Main characteristics (current section 5)
4.2. Comparison with previous similar work (current section 6.1)
4.3. Validation/Comparison of product with observational river discharge
     (current section 6.2)
4.4. Product uncertainties (current section 6.3)
```

**We have re-arranged the document into roughly the order that you suggest.**

2.2.2   2) Missing parts in method section

The method section should be extended – particularly the part about the derivation of the hydraulic characteristics. Specifically, I miss information about:

- How were (artificial) depressions in the DEM handled? With a filling algorithm?

**Correct re filling algorithm. This was not described in the submitted manuscript because the routing and filling algorithms are standard GIS tools. We've added a sentence explaining that sinks are filled so that all water is routed to leave the domain.**

- I'm confused about the applied flow direction algorithm. Was a single flow direction (SFD) or a multiple flow direction (MFD) algorithm used? And how were the basins delineated if a MFD algorithm was used (which has a dispersive character)?

**We used SFD-8.**

Moreover, the method used for assessing the basin uncertainty (section 6.3.1) should be moved to this section. It should include a more detailed discussion of the equation used to compute the hydraulic head and how this equation is applied to derive the sensitivity experiments in the appendix (with various subglacial pressure).

**We now only perform subglacial not supraglacial routing. Discussion about sensitivity experiment is now re-written into methods. Results are now discussed in result section, not appendix.**

2.2.3   3) Sensitivity of basins delineation to uncertainties in surface elevation and partitioning of surface/subsurface runoff

The evaluation of the Kangerlussuaq / Watson river catchment with river discharge data reveals that the accurate basin delineation is crucial. The sensitivity experiments with a different DEM for the surface and the consideration of subglacial drainage are thus extremely interesting and useful. I wonder if the uncertainty in the basin delineation, which is illustrated in the appendix, could be translated to runoff uncertainties (and be included in the runoff output product). One could for instance compute discharge at all (coastal) outlets for all sensitivity experiments and check the range in obtained runoff. This work would reveal catchments for which runoff quantifications are more (un)certain. It's probably not necessary to include these runoff uncertainty values in the current product but it would be nice upgrade.

**We have moved this sensitivity section into results rather than appendix. Unfortunately even these sensitivity experiments do not capture what we believe to be the "true" Watson basin (see Lindbäck et al. (2014) and Lindbäck et al. (2015)). We now manually select those "southern" basins to show that when included the modeled runoff matches the observed.**

K. D. Mankoff

We now provide subglacially routed water (for $k = 1.0$) rather than supraglacially routed water, as this better reflects reality. We generate subglacial routing for $k \in [0.8, 0.9, and 1.0]$ and use that for the sensitivity experiment. We also created a revision with $k \in [0.70.8, 0.9, 1.0, 1.09]$ (where $k = 1.09$ is effectively surface routing) but the paper became too complicated covering all those edge cases, 0.7 and 1.09 are extreme values, and releasing even 3 product for $k \in [0.8, 0.9, 1.0]$ seems unhelpful to end-users. We opted instead to briefly use the 3 $k$ scenario for the sensitivity experiment, but only discuss results from $k = 1.0$ and release that data.

As for using this simple sensitivity experiment to quantify runoff uncertainty - we are not sure how to do that. In the expanded Uncertainty section we now discuss the complexity of quantitatively estimating runoff uncertainty and how the basin uncertainty is not directly related, and even defining or comparing ice basins between $k$ scenario is difficult. For land runoff, the outlet is fixed. As you suggest we can and do take different $k$ simulations for upstream ice, route to a fixed land outlet, and look at the range of runoff from that outlet. The difference between $k = 0.8$ and $k = 1.0$ is minor - much less than the differences between RCMs, or an RCM and observations. Doing this for ice outlets is significantly more complicated because the basins and outlets change for each $k$ simulation, so it is not clear what should be compared between simulations.

It is a tractable problem to do manually for one or a few outlets. We provide the streams, outlets, and basins for $k \in [0.8, 0.9, 1.0]$, so that users can see possible changes in basin size, but only runoff for $k = 1.0$ to make the end-product simpler to use.

**2.3 Minor Comments**

**2.3.1 Content-related (text)**

Page 1 line 10: "contributes an additional ~35% to the ice runoff " → confusingly stated (because the ~35% are referring to the total runoff I guess) → rephrase

We have removed this text and the discussion between ice and land runoff.

P2L26-28: I don't understand to what "satellite basemap imagery" is referring to. To the ocean mask?

This was referring to the background satellite image in the map graphics of the basins for each observational data set. I have removed this text from this section of the document, but keep the "Basemap from Howat et al. (2014); Howat (2017a)" in each figure where the data is used, as per ESSD policy. I disagree with this policy.

P2L29: Mention somewhere here that RACMO only provides runoff for the glaciate area of Greenland

RACMO now includes land runoff, expands back in time to 1958, and both RACMO and MAR forward in time to 2019.

P3L5-7: The runoff downscaling should be explained in more detail (or a reference for the procedure should be provided)

References added.

K. D. Mankoff

P3L8: Is it justified to assume that the firn layers in both simulations (MAR and RACMO) are in approximate equilibrium in 1979 (i.e. was there a spin-up performed or when did the simulations start?)

**Yes, spin-up occurred prior to results being provided here.**

P3L14-17: How are (artificial) inland depression treated that would lead to erroneous inland outlets. Are such depressions apparent in the DEM? And if so, how are they removed?

**All depressions are filled until runoff leaves the domain (ice margin for ice runoff, coast for land runoff). This is now clarified in the text.**

P3L15: I'm confused by the part "multi-flow direction from eight neighbors". Does it imply that a multiflow direction algorithm with dispersion was used? Or a D8 algorithm (because this algorithm also allows flow from (maximal) eight neighbors).

**Multi-flow can come in from 8 neighbors, but only leaves to one. The text has been clarified.**

P3L26-27: I'm not sure if I understand this sentence correctly: so land pixels surrounded by ice are set to ice (but their elevation is left unchanged)?

**Correct. These local bumps may impact local streams, but should have no other effect because results are reported at the outlet, and the bumps are internal to basins, or if they define basin boundaries they still do even if their classification has changed. Put differently, inland ice streams often terminate (incorrectly, we presume) at nunatuks. We therefore set nunatuks to type 'ice' for the routing so that everything is routed to the ice margin. We then reclassify as 'land' so that the ice basin is not artificially enlarged for the runoff estimates.**

**We now explicitly point out that streams in the ice domain are merely "representative" of the model streams, but do not likely reflect actual subglacial streams, unlike the land streams which do appear to follow actual streams when compared against satellite imagery.**

P4L18-20: I don't understand this part: Is the downscaled gridded runoff data provided on an EPSG:3413 map projection (because I guess the direct output from the RCMs is on a rotated lon/lat-grid)? And the EPSG:3413 projection is based on WGS 84 (and thus an ellipsoid). But some data is provided in a coordinate system based on a sphere (earth spheroid)?

**I'm not entirely sure about the internal model grids - some are on a rotated pole lon/lat grid. They are provided to me on a EPSG:3413 projection. In EPSG:3413 1 $m^2$ is not equal to 1 $m^2$ in the real world, because of projection errors. We scale the data to account for this scale effect.**

P4L22-23: It should be stated in this section that land quantities (e.g. basin polygons and runoff) also include the same quantity from the glaciated part (I assume). So I guess runoff from land contains both runoff from the unglaciated and the glaciated part?

**The land runoff only include runoff that originated on the non-ice-covered land. In the data set you saw with the initial submission, land polygons included the**

**upstream ice area. In the current version we provide both the polygons with the upstream ice area, and the polygons cropped to only the ice-free land area, which is where the RCM land runoff is partitioned.**

**Including the land basins with upstream (under-ice) included is useful so that a point placed on the ice can easily determine which land basin contains the land outlet.**

P5L8-10: Why are the more larger land basins than ice basins? Do the land basins incorporate the ice basins?

**Yes - in the first version the land basins incorporated the ice basins, but land runoff was only calculated from the MAR land cells over the ice-free land portion of those basins. Land basins have to include the ice basins at some point in the processing because otherwise the routing algorithm will treat the ice edge as the edge of the domain, route streams there, and place outlets there. Having the land domain include the ice area forces outlets to the ocean boundary.**

**In this revised version we now crop the land domains to the ice-free portion after the routing algorithm step.**

P5L20-22: It should be more clearly stated in this sentence that the 4380 m3 refers to runoff from a single basin.

**Sentence removed.**

P5L24: I assume the ±30 km3 represent the RCM runoff uncertainty of 15% (this should be clearly stated here). And shouldn't it rather be ±60 km3? And how is this value of 15% derived (is there a reference)? I think it would be useful to mention this uncertainty value already in section 2 (input data).

**This sentence removed at the suggestion of reviewers. Annual runoff is not the point of this work. But we did add a sentence about RCM uncertainty when we introduce the data. You are correct that errors of X % should have been ± X, not ± (X/2)**

P6L8-11: This sentence does not belong here but rather in section 6.3.1. Furthermore, I find the sentence a bit hard to understand (particularly the last part) – could it be rephrased? It states that flow-path derived from the ArcticDEM generally agree better with satellite images than flow-path derived from BedMachine data, right?

**Correct, clarified, and moved.**

P6L17: Could you explain the reason why the increase in spatio-temporal resolution increases the signal-to-noise ratio in more detail? And I would include a reference to section 6.3.4 here (so that the reader knows where this strategy is discussed in the manuscript).

**More detail added and reference to Mitigation section where it is discussed even more.**

P6L18-26: I would move this part to the data section (2).

**Done - and in table form.**

P7L33-34: "MAR runoff slightly overestimates the GEM observations early in the year, and slightly underestimates the observation late in the year" → this is an interesting finding and probably related to storage of water in the (un-)glaciated area of the basin on intraannual time scales

**It may be interesting, but it is no longer in the revised manuscript. Given the additional observational data sets we now show all data from only the last year of each data set. This feature is no longer apparent and is not discussed in the revision.**

**MAR has been updated (v3.9 to 3.11) and I don't think this interesting artefact is there but not being show. It would show up in the scatter plots if it exists, and I don't see it there (consistently).**

P8L3-4: This step-like change in MAR runoff is rather strange. Are you certain that this is not an artefact (e.g. caused by an issue in MAR, the statistical downscaling procedure of runoff or the alignment of the 1 km and the 100 m masks)?

**We no longer see this feature. We are using now MAR 3.11 instead of 3.9, which may also be one reason it has disappeared.**

P8L27: "slight lag between models signals and the observations." → could this time lag be related to the neglect of routing travel time?

**Perhaps. We now focus more on bulk analysis, and use what we think are more appropriate graphics than a time-series, although the time series is still included.**

P8L28-29: What is the reason for the significantly higher temporal variability in RACMO? Could this be linked to the different treatment of liquid water retention on bare ice between the RCMs?

**In the revised manuscript MAR has been updated from 3.9 to 3.11, and RACMO now includes land runoff. This artefact is no longer present.**

P9L1-5: Why was the existing proxy data not used for further model validation (if it exists)?

**We have remove this text. This proxy data is a bit far removed from the models to be useful for validating them - turbulent plumes exist and must be modeled. The more likely scenario is that this product released here is used as inputs to those downstream studies. Indeed, that is what has happened in the past, but each study needed to do the work done here, wasting effort.**

P9L12: There is no equation 1

**There is, but we only referred to it immediately around it, and therefore did not reference the number (1). That has been changed.**

P9L15: "because large volumes of runoff usually come from large areas." → I do not understand this part of the sentence, could it be rephrased?

**"areas" should have been "basins". Sentence no longer in revised text.**

P10L4-5: What is meant by "hydraulic jumps"? I guess not the physical phenomena in hydraulics. If not, this term should be replaced to avoid ambiguity.

**That is the precise meaning. This occurs when masks are misaligned. Whenever the Citterio et al. (2013) ice mask transitions from ice to land, if BedMachine ice thickness is not 0, then the system transitions from subglacial to land surface in an abrupt fashion. There will be something unrealistic - either a waterfall or a sink that needs to be filled and may flow out somewhere else. There are many small basins along the coast and ice margin and many of these are realistic, but some may be due to the mask issues described above. This is discussed in more detail in the revised text.**

P10L11: This equation (and the corresponding text) should be moved to the method section.

**Now rewritten because we only use subglacial routing. This equation and the description of subglacial routing is moved up to the Methods section.**

P10L17-19: I find the transition between the previous and this part a bit strange. The part before explains how routing and basin delineation is derived when bed elevation is considered (this part should anyway be explained in the methods section in my opinion) but this section compares basin delineation based on two different surface topographies.

**This text has been removed.**

P10L30: Can you provide a reference for this value of 15%? Also, this value should already be mentioned in the data description (section 2).

**Additional text and reference added. Value introduced in the Methods section. More text in the RCM Uncertainty section.**

P11L4-5: Replace "highlighted above" with reference to relevant section. Additionally, are you certain that the step-like changes in RCM runoff originates from the actual RCM simulation (and is not generated by the subsequent postprocessing steps (e.g. downscaling or grid cell alignment).

**Text removed.**

P11L15: "current limitation" → future RCM simulation will still only capture features and process of certain spatial scales. But do you think that the most crucial scale will be represented in these simulations with higher resolution?

**Text removed, but yes, I think when the RCMs are run at 1 km or 100 m they'll do well enough. I think they do already (depending on your use case) after seeing the agreement between RCM and observations here.**

P12L4-5: Can you provide a reference that supports this (net storage is approximately zero) assumption?

**I cannot, but have made the statement less certain.**

P12L21-23: This sentence should be rephrased or removed. Making a prediction about fjord precipitation from the Greenland-wide fraction of land runoff is not reasonable in my opinion.

**Removed.**

P13L6-7: "perhaps due to temporal directionality" → I don't understand this part

**Removed. Time moves in one direction (?) so we cannot cite papers that have not yet been written, or document bugs we have not yet found. We use the GitHub website for this work to note those papers and issue that arise after publication.**

P13L7: Is "version of the dataset" meant here?

**Both. The document, code, and dataset from that code are all versioned to the same GitHub hash. Parts of this document write itself, using the data it generated. The NetCDF files also have the git hash. We prefer to leave the sentence as-is. The document has a git hash. The dataset has a git-hash for an earlier version of the document. If you compare changes between the dataset has and the current document hash, changes only occur in the text portions of the document (that are exported for the journal to publish) not the code portions of the document that generated the data set, or the code changes don't impact the data values (for example, only changes to metadata or file location).**

P13L20: Again, are the stated uncertainty values correct?

**We've doubled the errors - they are now ± 15 %, not ± 7.5 %. Not shown, but discussed, when time series plots show the ± 15 % from all three $k$ scenario, differences in runoff from $k$ are « differences between RCMs or between RCM and observations.**

2.3.2  Typos, phrasing and stylistic comments

Page 1 line 10: Change "over the time series" to "over time"

**Sentence changed.**

P2L4-5: I don't understand the meaning of this sentence ("Immediately upstream from…") – isn't it obvious that no submarine melting occurs upstream of the grounding line?

**Sentence changed.**

P3L12-13: "Each outlet has one upstream basin and each basin has one outlet" → I don't understand the meaning of this sentence, isn't this fact obvious?

**Removed.**

P3L22: Change "100 m2 pixel" to "10,000 m2 pixel"

**Change to "100 m x 100 m pixel"**

P4L12-13: "In the case of a small basin," → this sentence is a bit oddly stated – could it be rephrased?

**Rephrased.**

P4L30-31: I would remove "four per year" (and optionally change "provided as annual NetCDF files" to "provided as four annual NetCDF files").

**Done.**

P5L6-7: "Runoff ice products..." → oddly stated sentence → rephrase

**Rephrased. Also, we now use "discharge" to refer to our product, to keep it distinct from the source of the water, which is runoff from the RCMs.**

P5L21: "2012-08-06" → write date out

**Sentence removed.**

P5L27: "contributes an additional 35% to the ice runoff" → again, I find this a bit confusingly stated (I guess the 35% refer to total liquid runoff?). Maybe better: "contributes 35% to total runoff"

**Sentence removed.**

P6L5: Maybe change "and additional data products." to "and is provided with additional data."

**Done.**

P6L23: change "results to all observations that we have been able to find that are publicly accessible" to "results to all publicly accessible observations we could find"

**Done.**

P7L1: change "with high melt or runoff; Basin" to "with high runoff (and associated melt): Basin"

**Done.**

P7L7: change "include ice to the south of itself" to "include a glaciated area to the south"

**Reprhased.**

P7L23: "and a without an ice basin does have RCM ice cells" → odd formulation → rephrase (e.g. change "without an ice basin" to "unglaciated"

**Rephrased.**

P8L11-12: Rephrase sentence, e.g. to "The MAR relative runoff bias ranges from -20% (last day of time series) to +140% (28 July)."

**Text removed, but we now discuss biases much more extensively.**

P8L25: change "models than the observations" to "models than in the observations"

**Text removed.**

P8[edit: 9]L7: change "discussed below" to "still discussed in Sect. 6.3.2."

**Done.**

P8[edit: 9]L8: change "source uncertainty – the routing model, which exhibits in two different ways: Spatial (basin delineation) and temporal (runoff delay)" to "source of uncertainty – the routing model, which generates both spatial (basin delineation) and temporal (runoff delay) uncertainty"

**Done.**

P9L10-11: Rephrase to e.g.: "Temporal uncertainty is not systematically addressed in this work but a method to reduce it is discussed in Sect. 6.3.4."

**Rephrased.**

P10L19-20: Change sentence to e.g. "Results from additional sensitivity experiments (with different input data and hydraulic head computations) are shown in the Appendix."

**This section rewritten and moved.**

P10L29: rephrase "they do not precisely nor accurately capture reality" to e.g. "they represent reality discretised and simplified."

**Done.**

P12L11: change "That ice downstream" to "The downstream ice"

**Done.**

P12L26: Change "are approximately steady state" to "are approximately in steady state"

**Done.**

P12L30: Replace "GIS-wide ice sheet surface runoff" with "Greenland-wide ice sheet surface runoff". Otherwise, "GIS" is used both for "Greenland Ice Sheet" and "Geographical Information System"

**Sentence removed, but we are more careful with our use of GIS.**

P13L2: Replace "This work in its entirety is available" with "Output data of this work and part of the discharge observations are available"

**Rephrased.**

P13L8-9: This sentence is a bit oddly stated. Could you rephrase it?

**Paragraph removed.**

P13L12-15: This sentence is rather complicated to read and understand. Could you rephrase it?

**Paragraph removed.**

P13L15: change "differences in needed" to "differences are needed"

**Paragraph removed.**

P13L21: change "displaying and overall increase in both magnitude and variability" to "an overall increase in both its magnitude and variability"

**Sentence removed.**

P13L22: change "scale" to "scales"

**Done.**

**2.3.3 Figures and Tables**

Figure 1: Change caption to: "Overview showing ice basins (blue), land basins (green) and locations of following map figures (black)."

**Figure modified and caption changed.**

Figure 2: Change "Sec." to "Sect." in caption (also other occurrences)

**Figure removed, but we use "Sect." everywhere now.**

Figure 3: What is meant with "(this)" in the caption?

**Figure removed, but "this" was referring to this product, not 3rd party products.**

Figure 4: Maybe the error bars should be removed from this figure to improve readability. Additionally, "(this product based on ArcticDEM basins in Fig. 5)" should be rephrased.

**Figure simplified and combined with following figure.**

Figure 5: It's difficult to distinguish between different basins in this plot. Maybe readability could be improved by only plotting the basin's boundaries (without hatching). Could you also plot the Lindbäck et al. (2015) basin and the one you used to produce the right panel of figure 4?

**Figure simplified and combined with previous figure.**

Figure 6: A reference to this figure only appears in section 6.3.1, so its number and position should be changed accordingly. The legend is hard to read (it could be moved outside of the map area). Additionally, I would remove the sentence: "Region is zoomed in near Sermeq Kujalleq (Jakobshavn Isbræ)."

**Figure removed.**

Figure 7: Change "Fig. 10" to "Fig. 10 and 11" and "visible is basin artefact" to "visible is a basin artefact". Is the "RCM ice" mask showing the mask from RACMO or from MAR (also in the following figures)?

**Figure edited to include runoff time series and scatter plot in other panels. RCM mask is now the same for RACMO and MAR.**

Figure 8: Use "Fig." or "figure" consistently.

**Done.**

Figure 10: Change "Only 2017 shown" to "Only 2017 is shown"

**Text changed.**

Figure 11: Again, I would remove error bars to increase readability.

**Done.**

Figure 13: Change "Uncertainty only shown for total MAR runoff, not ice or land components." to "Uncertainties are only shown for MAR total runoff and not the individual land/ice components."

**Figure removed. Simplified figures now show total runoff, not separating land and ice.**

Figure 15: This plot is very hard to read. Again, I would remove the error bars. It also difficult to distinguish MAR from RACMO. Additionally, "MAR" should be removed from the y-axis labelling.

**Figures changed.**

**2.3.4 Supplementary Material**

Figure B1: Remove "not zoomed in". Additionally, I would always provide all necessary information in the figure caption about the comparison (experiment setting, margin/coastal outlet). References to other figures implies constant switching between figures. This also applies for the following figures.

**This appendix and these figures have been removed and incorporated into a single figure in the main text.**

**However, we've gone in the opposite direction regarding figure captions, and they are now even briefer and send the reader elsewhere - not to another figure, but to a section of the text. We know this is not ideal but really don't know how else to handle the current situation. There are 10 figures that are nearly identical, all with six panels. Repeating information for each figure seems unnecessary, and given the height of the figures, there may not even be space. Each figure caption would be 10s of lines. We hope you are OK with the modified figure captions referring readers the Methods section where we introduce the six-panel layout in full detail.**

**2.4 Review of provided dataset**

The presented dataset provides, to my knowledge, a unique and new source for high-resolution discharge data for the GrIS and the unglaciated part of Greenland for the present-day climate (1979 – 2017). The dataset seems very useful for downstream application in various field like e.g. hydrology, ecology and oceanography (particularly for fjord systems). The dataset can be accessed via the provided link, is complete and sufficiently supported with metadata and seems to be of good quality.

However, the description of certain processing steps is insufficient in my opinion and should be improved in the manuscript (see point 2 under "Major comments").

**We're happy to hear you think the data set is important and well-produced. The manuscript has been improved with your help and we hope it is now a better document in support of the data.**

**2.5 Minor Issues**

I was not able to found units for the Qaanaaq discharge dataset (https://promice.org/PromiceDataPortal/api/download/0f9dc69b-2e3c-43a2-a928- 36fbb88d7433/version_01/mel )

**https://doi.org/10.22008/hokkaido/data/meltwater_discharge/qaanaaq works for me now. I'm not sure what the issue was with the URL you used. Regardless, that DOI remains but GEUS has updated things are we are now using more formal software (the Harvard Dataverse) to server our data.**

When checking the static data (basin polygons, outlet locations and streams), I found some inland outlet locations near Kangerlussuaq. What is the reason for this?

**This was due to a mask issue and exporting an incorrect variable. It has been fixed.**

**3   Review 2**

This study provides high-resolution datasets of Greenland hydrologic outlets, basins, and streams, and a 1979 through 2017 time series of Greenland liquid water runoff for each outlet. This is a timely and important contribution for the Greenland hydrology community and I'm happy to see the paper and the associated datasets to be published. That said, I think some important issues need to be solved before it can be considered for publishing in ESSD.

**Timely and important are nice to hear. Your review was helpful and because of it the paper is now improved.**

**3.1   General Comments**

1/ The result section does not highlight the main contribution of this study very well. It includes numerous numbers of basins, outlets, streams, runoff but their importance is not well demonstrated. Furthermore, this section focuses on the total ice and land runoffs which can be easily derived from RCMs and have been well reported in previous studies. I suggest the result section should focus on what we can learn from runoff partitions in different basins, which is the new contribution of this study.

**Following your suggestion we have removed the annual and Greenland-wide comparison (we replace it with a monthly and Disko-only, highlighting the improvements in this work relative to Bamber et al. (2018). We have removed the results section at the suggestion of Reviewer 1 and now have a "Product description" section that highlights this new product.**

2/ The discussion section is too long and not easy to follow, particularly "6.2 Validation against observations". Most parts of section 6.2 should be removed to the result section. I suggest the authors only highlight the most important implications of their datasets and shorten this section.

**This section has been moved and rewritten, but it is not shorter. The previous Results section was 7 pages of text (3 validation, 4 uncertainty). It is now 11.5 pages (6 validation, 4.5 uncertainty, 1 summary). Importantly, we think the new layout may make it more efficient to read - the contents are more clearly broken down by section and if a reader wants to skip the detailed comparison between modeled discharge and observed discharge (after reading one or two and seeing the pattern), they can more easily do so.**

3/ It is important to mention that moulins are not identified so stream networks are delineated to continuously flow from inland to ice edge outlets. Therefore, the stream product may not represent the actual hydrological environments where moulins are widely distributed and fragment drainage networks, such as southwest GrIS. In contrast, the stream product may reasonably predict northwest GrIS drainage pattern since no moulins form there. Moreover, the contributing area threshold should be better illustrated since it determines the extent of streams. It may be useful to state

that the derived stream product aims to represent the general meltwater flow pattern rather than the actual spatial distribution of supraglacial rivers and streams.

**We now route subglacially, meaning water is assumed to immediately enter the subglacial system. Neglecting some surface flow (likely just a few km citetp:yang_2016_internally) is not likely to impact results because results are reported at the outlet. We also clarify that subglacial streams are model creations and do not represent real streams, although land streams appear to match streams seen in satellite imagery.**

4/ The quality of the main figures should be improved. Currently they are not satisfactory for publishing.

**Done.**

Also, the main point of each figure should be highlighted.

**We now refer readers to the section of the text where each figure is discussed. Please see detailed reply to Reviewer 1 about our figure captions, but briefly, given six panels it is not easy to briefly discuss one main point per figure, and often there isn't a specific point. The goal is to introduce data set users to the data, but given the diverse range of possible downstream users, it is difficult to know exactly what their use-case may be and therefore what their focus will be when trying to understand how the modeled discharge compares with observed. Furthermore, with six panels, figure captions will become lengthy, and with the figure repeating 10x, captions would be repetitive. We hope the current choice to limit figure captions and refer readers to the relevant text sections is acceptable.**

5/ More previous similar studies should be included. In the paper, the authors only compare their results with Lewis and Smith (2009). However, at least two important similar studies, Andersen et al (2015) and Pitcher et al (2016), should be added as comparison results.

**We spent some time searching for papers by Anderson in year 2015 (and other years), that mention Greenland. If you are referring to Andersen et al. (2015), we disagree that is a related paper. That paper (from a colleague at GEUS) addresses the multi-year mean surface mass balance to correct solid ice flux through "gates" at the PROMICE flight line at ~1700 m elevation. It is not about liquid water runoff, or anything at the daily resolution, or hydrologic basins.**

**Pitcher et al. (2016) is appropriate to cite and we thank the reviewer for reminding us about this paper. We now compare to it in the manuscript, but we are not sure it is "similar". It does address the uncertainty from the $k$ value, which we find to be less uncertain than the uncertainty introduced by multiple RCMs or observations. Beyond that, we do not see any other similarities - Pitcher et al. (2016) focus on one basin, do not provide any geospatial data, any time series of runoff, nor their code.**

**3.2 Minor Comments**

P1 L17, in this paragraph, I think it is necessary to say surface runoff contributes very importantly to Greenland mass balance (along with ice discharge).

**Done.**

P2 L4, it is not straightforward to understand "liquid runoff form surface melt, condensation, and rainfall".

**I'm not sure how to clarify this. We changed it to "liquid runoff from melted ice, rain, and condensation"**

P2 L22, why is 100 m ArcticDEM used to do the analysis?

**Elsewhere we clarify that when using the 150 m ArcticDEM, streams flow into the wrong fjord. ArcticDEM 100 m had no such artefacts. We did not see the need to use higher resolution ArcticDEM products.**

P3 L5, it may be worthy to mention that weathering crust of bare ice layer can store meltwater. Citation is required for this sentence.

**We leave this sentence as is. Many things can store water (tundra, cryoconite holes, crevasses etc.) but here we are not listing where storage could occur in reality, only pointing what level of storage are provided by the MAR and RACMO models.**

P3 L7, citation is required for this sentence.

**Reference added.**

P3 L10, it is not common to use the term "hydrologic head elevation".

**Changed to "subglacial pressure head", which we do see used in other literature (e.g. Gulley et al. (2014)).**

P3 L11, it is unclear how outlets are determined.

**Sentence removed. Outlets are provided by the 3rd party GIS tools we used for the routing.**

P3 L13, it is unclear what "major streams" means, some specific channel initiation thresholds (i.e. contributing area thresholds) are used to extract streams? It may be useful to call these "major streams" as rivers.

**Text has been clarified - we now add "above an upstream contributing area threshold", but keep the word "stream" throughout the document.**

P3 L17, why is 1 km2 used as threshold to merge small basins?

**It seemed an appropriate balance between too many micro-basins and generating too many unrealistic basins by combining larger basins that should be hydrologically distinct.**

P3 L20, "When this value is negative, it indicates submarine (subglacial) discharge", this sentence is not clear.

**Sentence has been removed.**

P3 L21, this section is too long. I suggest it should be shorten or some parts can be put into SUPP.

**We have shortened the section as per your suggestion.**

P4 L24, see my general comment, more explanations should be provided for the stream product.

**Done.**

P5 L8, it is not easy to understand what these numbers mean and why they are important

**Paragraph removed.**

P5 L15, what does "adjusts" mean here?

**This paragraph has been rewritten.**

P5 L21, 4380 m3 rather than 4380 m-3.

**Paragraph removed.**

P5 L23, which basin? also report the similar value in Lewis and Smith (2009).

**Paragraph removed.**

P5 L26, Mt. Pinatubo eruption, add a citation to support this result.

**Paragraph removed.**

P5 L27, the land runoff is considerably large. It is useful to further illustrate its meaning.

**Paragraph removed.**

P6 L7, "Routing with a 5 km DEM is likely to cause some basins and outlets to drain into an incorrect fjord", what is the reason for this?

**Changed to "Routing with a 5 km DEM that does not capture small-scale topography". A lower resolution DEM may miss a hill or small mountain range that changes modeled stream patterns.**

Fig 2 is not easy to understand. What is the main point of this figure? Perhaps remove it to the SUPP?

**It was a graphical depiction of the coverage issues. Removed.**

Fig 4, it is not clear why runoff from the Watson River basin plus the two large basins immediately to the south performs better.

**This section has been re-written. The reason is likely that the true basin includes the two large basins to the south - that is what is contributes to the observed runoff, so that is what should be included in the model.**

Insets are required to show the location of Fig 5 -9 in Greenland.

**These figures have been changed but do not include an inset. We invite the reader to refer to Figure 1 for location.**

Fig 6 is not easy to follow. What is the meaning to change outlet locations?

**The figure has been changed (and is now Figure 2) and the text describing the methods has been clarified and made more central. It is now in the methods section, not in the supplemental material.**

Fig 7 is not easy to follow. What is the main point of this figure?

**Changed, but still included, and now repeated for each of the observation locations. These figures (now panel a for each location) provide an overview of the observational field site and environment, the basins (land and ice) and the RCM coverage.**

Merge Figure B1-B8 into one figure.

**Done (now Figure 2).**

**4   References**

Andersen, M. L., L. Stenseng, H. Skourup, W. Colgan, S. A. Khan, S. S. Kristensen, S. B. Andersen, J. E. Box, A. P. Ahlstrøm, X. Fettweis, and R. Forsberg (2015). "Basin-scale partitioning of Greenland ice sheet mass balance components (2007 – 2011)". *Earth and Planetary Science Letters*. 409, 89–95. DOI: 10.1016/j.epsl.2014.10.015.

Bamber, J. L., R. M. Westaway, B. Marzeion, and B. Wouters (2018). "The land ice contribution to sea level during the satellite era". *Environmental Research Letters*. 13 (6), 063008. ISSN: 1748-9326. DOI: 10.1088/1748-9326/aac2f0.

Citterio, M. and A. P. Ahlstrøm (2013). "Brief communication "The aerophotogrammetric map of Greenland ice masses"". *The Cryosphere*. 7 (2), 445–449. ISSN: 1994-0424. DOI: 10.5194/tc-7-445-2013.

Gulley, J. D., P. D. Spellman, M. D. Covington, J. B. Martin, D. I. Benn, and G. A. Catania (2014). "Large values of hydraulic roughness in subglacial conduits during conduit enlargement: Implications for modeling conduit evolution". *Earth Surface Processes and Landforms*. 39 (3), 296–310. DOI: 10.1002/esp.3447.

Lindbäck, K., R. Pettersson, S. H. Doyle, C. Helanow, P. Jansson, S. S. Kristensen, L. Stenseng, R. Forsberg, and A. L. Hubbard (2014). "High-resolution ice thickness and bed topography of a land-terminating section of the Greenland Ice Sheet". *Earth System Science Data*. 6 (2), 331–338. ISSN: 1866-3516. DOI: 10.5194/essd-6-331-2014.

Lindbäck, K., R. Pettersson, A. L. Hubbard, S. H. Doyle, D. van As, A. B. Mikkelsen, and A. A. Fitzpatrick (2015). "Subglacial water drainage, storage, and piracy beneath the Greenland Ice Sheet". *Geophysical Research Letters*. DOI: 10.1002/2015GL065393.

Pitcher, L. H., L. C. Smith, C. J. Gleason, and K. Yang (2016). "CryoSheds: a GIS modeling framework for delineating land-ice watersheds for the Greenland Ice Sheet". *GIScience & Remote Sensing*. 53 (6), 707–722. ISSN: 1943-7226. DOI: 10.1080/15481603.2016.1230084.

---

## Referee Report (RR1)

**Summary and general comments**

I would like to thank the authors for the substantial amount of work they put in the revision of the manuscript. The structure was much improved. I have a couple of minor additional comments (the page and line numbers refer to the latest manuscript version). The comments often concern sentences that are unclearly worded and hard to understand.

**Content-related (text)**
**Line 2:** "where and when" → it's maybe better to leave out the "when" here. The timing of discharge is rather uncertain (→ runoff routing delay was not accounted for in this product).
**Line 6:** shouldn't it be 22645 days?
**Lines 10 – 12:** "spanning four orders of magnitude … +500%/-80%). → this part is difficult to understand (without reading the manuscript). Can you rephrase it?
**Line 26:** what is meant by "stream spatial resolution"?
**Line 42:** I would already mention "bed topography" here
**Line 57:** "ERA 6-hour" → I guess "ERA-Interim 6-hour"
**Line 57:** "RACMO (v 2.3p2; Noël et al. (2018)) ran with 5.5 km" → is it really 5.5 km – not 11 km?
**Line 65:** what is meant by "other RCM output"?
**Line 76:** I find the term "±95 % quantile range" odd. An alternative name could be "5-95% quantile range"
**Line 93:** what values was selected for this threshold?
**Line 107:** I was not able to find the "seven-day smoothing filter" in Van As et al. (2017)
**Lines 122 – 125:** It's difficult to follow this (long) sentence – could you rephrase it?
**Line 126:** I guess it should rather be "map projection of the statistically downscaled RCM product" instead of "map projection of the RCM"
**Line 126 – 128:** This part already confused me in the previous manuscript version. However, I think I understand it now: You have to perform the scaling because EPSG:3413 is not an equal area map projection, right?
**Line 141:** "The RCM ice domain" → is the MAR or RACMO ice domain shown?
**Line 149:** "the 95% prediction interval" → how to you compute this interval exactly?
**Lines 160 – 161:** I don't understand this sentence: it seems odd to assume subglacial flow but compare streams with supraglacial features.
**Line 166:** remove "Alternatively" (also in line 169)
**Line 202 – 203:** "That runoff is both…" → I don't understand this sentence; could you rephrase it?
**Line 211:** Why is the performance of MAR (0.45) much lower than the one of RACMO (0.88)?
**Lines 212 – 213:** "For RACMO this is…" → could you rephrase this sentence?
**Lines 219 – 220:** replace "not necessarily…" by "not necessarily the insufficient ability of the RCMs to simulate (near-surface) climate conditions."
**Line 222:** "for all and only the days" → I don't understand
**Lines 222 – 223:** "for example the…" → oddly stated, please rephrase
**Line 224:** "reports ~50% of the observed discharge" → not really visible from figure 4
**Line 226:** "where the RCMs do not cover…" → does that apply both to MAR and RACMO?
**Lines 232 – 233:** "or half of the range of the data." → I do not understand this part
**Lines 264 – 266:** could you rephrase this sentence?
**Line 272:** "There is no way…" → I don't understand the meaning of this sentence.
**Lines 274 – 275:** "The other two…" → I don't understand this sentence
**Lines 285 – 287:** I find it a bit odd that the gauge location is shifted onto the ice. Can you explain this choice in more detail?
**Line 307:** what does ENE mean? East-Northeast? (same for "NNE" a line below)
**Lines 327 – 328:** "These agreements…" → I don't understand this sentence
**Lines 350 – 351:** "Any ArcticDEM…" → could you rephrase this sentence?
**Line 374:** what is meant by "almost-overlapping ice basins"? Generally, this paragraph is difficult to understand (in my opinion) and could be improved.
**Lines 392 – 393:** " and the range of upstream…" → I don't understand this part
**Line 419:** "the relative uncertainty between the bed to the surface increases." → I don't understand
**Line 420:** "may overflow away" → I'm not sure what is meant by this
**Line 431:** is this sentence correct? → "examined the uncertainty of modelled SMB for 95%"
**Line 465:** what you mean by "coverage algorithm"?
**Line 466:** "discharge can be" → I would replace "can" by "could" (because this method is not applied in your work; right?)

**Line 469:** do you really apply a lag function in this work? I thought it is only a seven-day smoothing.
**Lines 484 – 485:** "pushing the coast into fjords…" → I don't understand this part
**Line 486:** "10 m bin at 0 m elevation." → I guess this refers to the bin ranging from 0 – 10 m, right?
**Line 486:** does "± 10 m" refer to 0 ± 10m?
**Line 500:** why only the "non-ice-covered land surface"?
**Line 503:** what is meant by "from all previous freshwater sources"?
**Line 513:** neglecting routing delay also contributes to the uncertainty in discharge timing and should be mentioned here.
**Line 515:** I don't understand "half an order-of-magnitude" here (also in line 580)
**Line 517:** again, do you really introduce a temporal lag?
**Line 526:** "and may be systematic (bias)." → what do you mean by that?
**Lines 528 – 530:** I think this statement is incorrect: the errors add up according to $e_{tot}^2 = e_1^2 + e_2^2 + …$, right?
**Line 587:** What do you mean by "process-level"?

**Typos, phrasing and stylistic comments**
**Line 54:** "RCM results" → "RCM output"
**Line 141:** ", and RCM land domain not shown" → "and the RCM land domain is not shown"
**Line 150:** "observations vs. difference" → observations vs. bias"
**Line 258:** change "van As et al. (2012)." to "(van As et al., 2012)
**Line 261:** remove "here"
**Line 274:** change "Uncertainty section" to "uncertainty section"
**Lines 390:** "… small enough it is usually difficult …" → transition within sentence could be improved
**Line 445:** I would remove either "precise" or "accurate"
**Lines 446 – 447:** replace "but not …" by "but not in the scatter plots."
**Line 453:** replace "equivalent many" by "equivalent to many"
**Line 474:** is "prohibitive" the correct word here? Maybe "intensive" is better…
**Line 519:** replace "to annual sum" by "to annual sums"
**Lines 583 – 584:** replace "through spatial or…" by "through spatial/temporal aggregation or by implementing a lag function."

**Figures and Tables**
**General comments:** The size of most graphics should be increased.
**Figure 5:** I'm confused by this graphic: why is the difference (bias) always positive? It should also be negative (if the observed value is higher than the modelled one), right? Or are the panels actually showing the relative biases (RCM (MAR or RACMO) / observation)? But then, the y-axis should be unitless.
**Figure 5:** caption: "RCM minus observations" → "RCM bias"
**Figure 7b:** I find it a bit confusing that only the last calendar year is shown here (but all data is used for the graphics below). Maybe it's better to put the full hydrographs in the supplementary material?
**Figure 7c:** the colour bar should be moved outside of the panel (for improved readability)
**Figure 18:** Are there really land outlets with elevations up to ~1500 m? And why are "absolute land outlet elevations" plotted in the bottom panel (and not also negative values)?

**Access to online data**
I briefly checked the "Discharge measurement at the outlet stream of Qaanaaq Glacier" and I'm still not able to find the units of the provided values. Are there stated somewhere?

---

## Author Response (AR2)

**Reply to Reviewers**

**Ken Mankoff *et al.**

Comments from reviewers are in normal font and differentiated from the replies that use a **bold colored font.**

**Contents**

**1 Reviewer 1**

I would like to thank the authors for the substantial amount of work they put in the revision of the manuscript. The structure was much improved. I have a couple of minor additional comments (the page and line numbers refer to the latest manuscript version). The comments often concern sentences that are unclearly worded and hard to understand.

**1.1 Content-related (text)**

Line 2: "where and when" → it's maybe better to leave out the "when" here. The timing of discharge is rather uncertain (→ runoff routing delay was not accounted for in this product).

**I don't want to mislead people, but I'd like to keep "when". There is a temporal component to the results. The timing is uncertain, but so is the location. Everything has uncertainty that we try to quantify here. We also recommend people downsample to weekly or monthly if appropriate, which is a "when".**

Line 6: shouldn't it be 22645 days?

**Yes, I computed it as (max-min), but it should have been (max-min)+1**

Lines 10 – 12: "spanning four orders of magnitude … +500%/-80%). → this part is difficult to understand (without reading the manuscript). Can you rephrase it?

**Changed: We compare RCM results with 10 gauges from streams with discharge rates spanning four orders of magnitude. Results show that for daily discharge at**

**individual basin scale the 5 to 95 % prediction interval between modeled discharge and observations generally falls within plus-or-minus a factor of five (half an order of magnitude, or +500%/-80%)**

Line 26: what is meant by "stream spatial resolution"?

**Changed: "high spatial resolution (~100 m; resolving individual streams)"**

Line 42: I would already mention "bed topography" here

**Added: an ice sheet bed DEM**

Line 57: "ERA 6-hour" → I guess "ERA-Interim 6-hour"

**MAR is ERA5 (added "5"). RACMO is ERA-Interim.**

Line 57: "RACMO (v 2.3p2; Noël et al. (2018)) ran with 5.5 km" → is it really 5.5 km – not 11 km?

**Yes it really is 5.5 km resolution**

Line 65: what is meant by "other RCM output"?

**Removed.**

Line 76: I find the term "±95 % quantile range" odd. An alternative name could be "5-95% quantile range"

**Yes - changed throughout.**

Line 93: what values was selected for this threshold?

**Added: (> 3 km$^2$)**

Line 107: I was not able to find the "seven-day smoothing filter" in Van As et al. (2017)

**Thank you for catching this. van As et al. (2017) uses 10 day smoothing. We originally used 10 day but changed to 7 day, and did not properly adjust the reference. Changed to "so all analyses done here include a seven-day smooth applied to the RCM discharge product (cf. van As et al. (2017))."**

Lines 122 – 125: It's difficult to follow this (long) sentence – could you rephrase it?

**Changed: Conversely, when misalignment is proportionally large (e.g. a basin is only 1 % covered by the same RCM classification), this implies a small basin. Because the basin is small, the covered region (no matter how much smaller) must be nearby and not climatically different.**

Line 126: I guess it should rather be "map projection of the statistically downscaled RCM product" instead of "map projection of the RCM"

**Yes, but as far as I know the downscaled product are on the same projection as the non-downscaled product. Also, I think it is implied that the term "RCM" throughout the document refers to our input data, which is statistically downscaled.**

Line 126 – 128: This part already confused me in the previous manuscript version. However, I think I understand it now: You have to perform the scaling because EPSG:3413 is not an equal area map projection, right?

**Yes! That is a better way to phrase it. Thank you. Changed to: "RCM inputs are also scaled to adjust for the EPSG:3413 non-equal-area projection. This error is up to..."**

Line 141: "The RCM ice domain" → is the MAR or RACMO ice domain shown?

**With the updated RCMs used in the revised work, both are now on the same domain. Added: Both MAR and RACMO use the same domain.**

Line 149: "the 95% prediction interval" → how to you compute this interval exactly?

**Ordinary least squares (OLS). "Exactly" can be seen in the code block here: `https://github.com/mankoff/freshwater/blob/d4e98e18c2425e0b652b640d01410494ae220acf/freshwater.org#scatter-at-each-obs`. More generally, I use `https://www.statsmodels.org/stable/generated/statsmodels.regression.linear_model.WLS.html#statsmodels.regression.linear_model.WLS` but with weights = 1, which should be equal to `https://www.statsmodels.org/stable/generated/statsmodels.regression.linear_model.OLS.html#statsmodels.regression.linear_model.OLS`.**

Lines 160 – 161: I don't understand this sentence: it seems odd to assume subglacial flow but compare streams with supraglacial features.

**I have nothing else to compare to. Also, from Eq. 2, the ice thickness controls ~92 % of subglacial head. If one assumes slopes of basal features are ~10x as large as surface slopes, this means that ice surface features still control ~50 % of subglacial routing. Furthermore, there is literature (cited in the next sentence) suggesting that the bed strongly influences surface features. I believe it is worth pointing out that in our case there may be two reasons for this agreement. As stated, (1) the processes described in the cited literature or (2) problems with BedMachine that mean what people think is subglacial routing is really supraglacial routing.**

Line 166: remove "Alternatively" (also in line 169)

**Done.**

Line 202 – 203: "That runoff is both..." → I don't understand this sentence; could you rephrase it?

**Removed. I think the preceding and following sentence are sufficient.**

Line 211: Why is the performance of MAR (0.45) much lower than the one of RACMO (0.88)?

**Unknown. A better question is why RACMO fairs better. From the graphic, MAR does not agree for the small basins where there are small glaciers in reality that are not included in the RCMS - coverage is 0. The question then may be why RACMO matches observed runoff when only simulating terrestrial runoff (rain and snow melt).**

Lines 212 – 213: "For RACMO this is..." → could you rephrase this sentence?

**Re-written.**

Lines 219 – 220: replace "not necessarily…" by "not necessarily the insufficient ability of the RCMs to simulate (near-surface) climate conditions."

**Done.**

Line 222: "for all and only the days" → I don't understand

**Clarified: "…for the days…"**

Lines 222 – 223: "for example the…" → oddly stated, please rephrase

**Rephrased.**

Line 224: "reports ~50% of the observed discharge" → not really visible from figure 4

**I disagree. The "W"s clearly cover the lower solid line, which is a 2:1 ratio.**

Line 226: "where the RCMs do not cover…" → does that apply both to MAR and RACMO?

**Yes.**

Lines 232 – 233: "or half of the range of the data." → I do not understand this part

**Added "(±25 %)". The data subset we describe here (top 2/3rds) spans 2 orders of magnitude. The uncertainty at ±0.5 order of magnitude spans 1 order of magnitude. 1 order of magnitude (uncertainty) is half of 2 orders of magnitude (range of data).**

Lines 264 – 266: could you rephrase this sentence?

**Simplified.**

Line 272: "There is no way…" → I don't understand the meaning of this sentence.

**Re-written.**

Lines 274 – 275: "The other two…" → I don't understand this sentence

**Added reference to previous section where we highlight three problematic areas. This sentence comes at the end of the first problematic area (Watson).**

Lines 285 – 287: I find it a bit odd that the gauge location is shifted onto the ice. Can you explain this choice in more detail?

**Added: (equivalent to selecting a different outlet)**

Line 307: what does ENE mean? East-Northeast? (same for "NNE" a line below)

**Yes. Expanded to full words.**

Lines 327 – 328: "These agreements…" → I don't understand this sentence

**Removed.**

Lines 350 – 351: "Any ArcticDEM…" → could you rephrase this sentence?

**Rephrased.**

Line 374: what is meant by "almost-overlapping ice basins"? Generally, this paragraph is difficult to understand (in my opinion) and could be improved.

**Re-written using what I hope is more clear language and examples.**

Lines 392 – 393: " and the range of upstream…" → I don't understand this part

**Clarified that the range comes from $k$ simulations. (I think it might have been interpreted as the range over time).**

Line 419: "the relative uncertainty between the bed to the surface increases." → I don't understand

**Rewritten: At the margin, many of the small basins (absorbed or not) may be incorrect because the bed uncertainty is larger relative to the ice thickness, and therefore uncertainty has a larger influence on routing.**

Line 420: "may overflow away" → I'm not sure what is meant by this

**Changed to: may overflow (i.e. the stream continues onward) somewhere at the sink edge different from the location of the real stream**

Line 431: is this sentence correct? → "examined the uncertainty of modelled SMB for 95%"

**Yes, this sentence is correct.**

Line 465: what you mean by "coverage algorithm"?

**Added reference to section where this is introduced.**

Line 466: "discharge can be" → I would replace "can" by "could" (because this method is not applied in your work; right?)

**Done. Correct, not applied.**

Line 469: do you really apply a lag function in this work? I thought it is only a seven-day smoothing.

**We do apply a smooth, but that is a form of lag.**

Lines 484 – 485: "pushing the coast into fjords…" → I don't understand this part

**Changed: placing a section of coastline in a fjord**

Line 486: "10 m bin at 0 m elevation." → I guess this refers to the bin ranging from 0 – 10 m, right?

**Correct. Changed: 0 - 10 m elevation bin**

Line 486: does "± 10 m" refer to 0 ± 10m?

**Yes. Clarified.**

Line 500: why only the "non-ice-covered land surface"?

**The flux to the surface should be the same for nearby surfaces regardless of if they are water, terrestrial surface, or ice surface. However, this product does not provide flux to, but rather flux from a surface. The flux from the ice surface has melt, and therefore cannot be used to estimate flux to the water surface. The flux to the water surface can be estimated by taking the discharge from any nearby land-only basin (where the discharge comes only from rain and snowfall), dividing by the area of that basin, and using that as an estimate of flux to the water surface.**

Line 503: what is meant by "from all previous freshwater sources"?

**Honestly I'm not sure what I was trying to say here. Changed: "from runoff".**

Line 513: neglecting routing delay also contributes to the uncertainty in discharge timing and should be mentioned here.

**Added: ", because routing delays are neglected"**

Line 515: I don't understand "half an order-of-magnitude" here (also in line 580)

**I'm really not sure how to clarify this. The previous part of the sentence is "plus-or-minus a factor of five". A factor of 10 is an order of magnitude. Half an order-of-magnitude = 1/2 * 10 * value = scaled by a factor of 5. Or +500 % / -80 %**

Line 517: again, do you really introduce a temporal lag?

**Yes. See example code below. The 10 is lagged because it is spread over the 3 following days.**

```
import numpy as np
import pandas as pd
np.random.seed(100)
df = pd.DataFrame(index=np.arange(10), data=np.random.random(10), columns=['data'])
df.loc[4] = 10
df['smooth'] = df['data'].rolling(4).mean()
df
```

|   | data | smooth |
|---|------|--------|
| 0 | 0.543405 | nan |
| 1 | 0.278369 | nan |
| 2 | 0.424518 | nan |
| 3 | 0.844776 | 0.522767 |
| 4 | 10 | 2.88692 |
| 5 | 0.121569 | 2.84772 |
| 6 | 0.670749 | 2.90927 |
| 7 | 0.825853 | 2.90454 |
| 8 | 0.136707 | 0.438719 |
| 9 | 0.575093 | 0.5521 |

Line 526: "and may be systematic (bias)." → what do you mean by that?

**Rewritten: For land basins, subglacial routing errors no longer exist, basins are well-defined, and errors are due to neglecting runoff delays or the RCM estimates of runoff.**

Lines 528 – 530: I think this statement is incorrect: the errors add up according to etot2 = e 12 + e22 + . . . , right?

**No that is not right. Using traditional uncertainty propagation and assuming random errors, errors grow by the square root of sum of the errors squared. That is, if your errors for two basins are 1 and 1, the sum of the two basins has an error of 1.414. I think this is a conservative estimate, as errors between neighboring basins are not random but are likely to be related and offset each other.**

Line 587: What do you mean by "process-level"?

**This is clarified by the existing text: "(i.e. glacier terminus for solid ice discharge, stream for liquid discharge)."**

1.2   Typos, phrasing and stylistic comments

Line 54: "RCM results" → "RCM output"

**Fixed.**

Line 141: ", and RCM land domain not shown" → "and the RCM land domain is not shown"

**Changed.**

Line 150: "observations vs. difference" → observations vs. bias"

**Changed.**

Line 258: change "van As et al. (2012)." to "(van As et al., 2012)

**Changed.**

Line 261: remove "here"

**Removed.**

Line 274: change "Uncertainty section" to "uncertainty section"

**Changed.**

Lines 390: ". . . small enough it is usually difficult . . . " → transition within sentence could be improved

**Clarified.**

Line 445: I would remove either "precise" or "accurate"

**These words precise and distinct differences. In this context, on represents systematic bias, and the other random error.**

Lines 446 – 447: replace "but not . . . " by "but not in the scatter plots."

**Done.**

Line 453: replace "equivalent many" by "equivalent to many"

**Added.**

Line 474: is "prohibitive" the correct word here? Maybe "intensive" is better. . .

**I believe this is the correct word.**

Line 519: replace "to annual sum" by "to annual sums"

**Done.**

Lines 583 – 584: replace "through spatial or..." by "through spatial/temporal aggregation or by implementing a lag function."

**Done.**

1.3   Figures and Tables

General comments: The size of most graphics should be increased.

**I will work with the copy editors for this.**

Figure 5: I'm confused by this graphic: why is the difference (bias) always positive? It should also be negative (if the observed value is higher than the modelled one), right? Or are the panels actually showing the relative biases (RCM (MAR or RACMO) / observation)? But then, the y-axis should be unitless.

**It is a ratio. I've re-labeled the y-axes.**

Figure 5: caption: "RCM minus observations" → "RCM bias"

**Done.**

Figure 7b: I find it a bit confusing that only the last calendar year is shown here (but all data is used for the graphics below). Maybe it's better to put the full hydrographs in the supplementary material?

**Our aim is not to confuse but to show representative data. Given that some stations have < 1 year of data and others have ~40 years, we opted to just show the last year. Anyone wanting the full hydrographs is encouraged to download the data which is freely available with a documented data access script.**

Figure 7c: the colour bar should be moved outside of the panel (for improved readability)

**Done.**

Figure 18: Are there really land outlets with elevations up to ~1500 m? And why are "absolute land outlet elevations" plotted in the bottom panel (and not also negative values)?

**There are in this data which is clearly incorrect. This is indicative of mask misalignment. We show absolute value because the errors near 0 are likely to be randomly distributed around 0 - less than 0 if the mask artificially places the coast ocean-ward of the true coast, and greater than 0 if the mask artificially places the coast land-ward of the true coast.**

1.4   Access to online data

I briefly checked the "Discharge measurement at the outlet stream of Qaanaaq Glacier" and I'm still not able to find the units of the provided values. Are there stated somewhere?

You're right, units are not provided. I have updated the README and data file headers. That dataset is now at v2. At the moment we're having trouble re-issuing new DOIs for this - the DataCite service has a bug. But I have a draft version of this updated dataset in our Dataverse and will publish the update as soon as I am able to do so.

**2 Reviewer 2**

Figure 2, "maximum possible distance between outlet locations for all cells" is not easy to follow. It is a little bit misleading to apply a distance to an entire basin. It will be useful to change the figure design, or just better explain the distance in the figure caption.

We have tried to clarify the sentence. We disagree that showing distance applied to an entire basin is misleading. On the contrary, we believe it is an important view that captures the change (or possible changes) in outlet cell location for each upstream cell. It provides information about which basins or regions are well defined and robust to changes in subglacial routing assumptions, and which are sensitive. Furthermore, for those that are sensitive, it provides an estimate of how sensitive and if that sensitivity matters.

Figure 7, how is Figure 7b generated? I recall MAR and RACMO runoff is close to observed Waston river discharge but in Figure 7b the latter is much larger than RCM runoffs.

This question is unclear.

Fig. 7b is generated by plotting the observed data from van As et al. (2018) and the discharge generated for the nearby outlet as derived by this work.

Do you recall something from this paper? Throughout it we explicitly state that Watson modeled runoff is ~50 % of observed. If you recall something from somewhere else, perhaps it is van As et al. (2018), but we discuss in detail why that paper shows agreement between modeled and observed and why this one does not. We need more information to fully answer this question.

Will it be better to put Figure 9 to 17 in the supplementary? or put some of the representative figures in the main text and the remaining figures in the supplementary? This is optional though.

The many validation figures were in the supplementary material in the first version, and moved to the main paper after the suggestion of earlier reviewers.

Why do MAR and RACMO perform so differently in Teqinngalip (Figure 13) and Zackenberg (Figure 16)? Is it because these two small basins cover large land areas, and thus lead to large uncertainties in runoff estimation?

Unknown. It may be related to small basins which have glaciers but the glaciers are not in the RCMs.

[revised manuscript text omitted]

---

## Author Response (AR3)

Dear Reinhard,

Thank you for the helpful technical comments. I have done most of them, and the few I have not I explain
why. Answers to all your comments are below.

Regards,

    Ken Mankoff

+ [X] Avoid line breaks between numbers and units (e.g. p1/l5) with "100~m" instead of "100 m".

I'm not actually writing LaTeX but something that exports and I don't have this level of control. I make
sure these breaks don't appear during the proofing.

+ [X] P3/l59 What is this uncertainty based on? (Alternatively, provide cross-link to section 4.3.3)

Added: (Sect. 4.3.3)

+ [X] State somewhere in the MS that elevation is referenced to "sea level" throughout (and not, for
example, WGS84).

Added: Both DEMs are referenced to the WGS84 ellipsoid.

+ [X] Figs 8, 9 – 17 add spatial scalebar in (a). Consider increasing line thickness (particularly the blue
line is sometimes hard to see).

Done

+ [X] Figs 9 – 17: consider removing the x-axis (which is same as in d). This may add some clarity and
reduce confusion from some data points below the x-axis (e.g. Fig 12)

Done

+ [X] Fig 18 is "Frequency" the correct y-label? I would feel better with "counts" or something like that.
Frequency, at least for me, should have units of "Hz".

Changed to "Count" and "Cumulative counts"

+ [X] Reconsider the wording of "resolution" in conclusion and elsewhere. For example, the 100 m gridding of
the ArcticDEM was a choice of the gridding algorithm and may or may not reflect the "spatial resolution" of
the sensor applied. Often, I think, the word "gridding" is more adequate, but I leave this open to the
authors.

I've changed 'resolution' to 'gridded' (or similar) in many places in the text. I've opted to keep it as
'resolution' in the conclusion because I think it helps with sentence clarity, because the '100 m spatial
resolution' is paired with '1 day temporal resolution'.

+ [X] Acknowledgements: Consider thanking the reviewers. I think they did a commendable job.

Added (citations to the ESSD comments via DOI): The editor and two anonymous reviewers provided valuable
feedback and helped improve this paper (Anonymous, 2020a, b)